# ONE POLICY LEARNS THEM ALL: SYNERGIZING PRIOR-GUIDED EXPLOITATION AND ONLINE EXPLORATION IN CURRICULUM BASED MARL

## ABSTRACT

Existing offline-to-online (O2O) multi-agent reinforcement learning (MARL) methods typically employ offline prior policies for warm-start initialization but are susceptible to distributional shifts and structural consistency constraints. On the other hand, prior-guided cold-start conditions, albeit of more practical interest, require a subtle synergy between utilizing prior-collected samples and self-exploring the state-action space. In this paper, we propose **DUCE**, a **du**al-track **c**urriculum MARL algorithm that balances **e**xploitation and **e**xploration to ensure efficient, stable cold-start training. The curriculum designs include: (1) an externally configured task-difficulty curriculum that alternates between performing prior and online policies with probabilistic scheduling, progressively reducing the prior-guidance horizon to transition tasks from easy to hard, and (2) an internally evolving policy optimization curriculum that imposes a decaying offline RL regularizer on the online loss, enabling a smooth shift from conservative prior reliance to exploration-driven training. Extensive experiments on challenging StarCraft multi-agent challenge (SMAC) v1/v2 tasks demonstrate that DUCE achieves faster convergence and higher asymptotic performance, and consistently outperforms state-of-the-art warm-start baselines. Importantly, DUCE is agnostic to the architectures of priors (e.g., rule-based or RNN).

## 1 INTRODUCTION

Multi-agent reinforcement learning (MARL) is a core subfield of machine learning exhibiting notable research progress across diverse domains (Ma et al., 2024; Lai et al., 2025). Nevertheless, its applications in real-world scenes remain hindered by the curse of dimensionality in joint state-action spaces, the sparsity of rewarding experience and the non-stationarity arising from concurrently adapting agents (Li et al., 2025). These challenges typically lead to lower sample efficiency and less stable training compared with single-agent reinforcement learning (SARL) (Sunehag et al., 2017; Rashid et al., 2018; Shen et al., 2022).

To improve training efficiency, recent work (Zhong et al., 2025; Zhang et al., 2024; Zhou et al., 2025) have extended offline-to-online (O2O) methods from SARL to MARL, leveraging offline-trained policies as a **warm start** of online training. However, as shown in Fig. 1 (right), these approaches often suffer an initial performance drop and even finally underperform baselines without initialization, due to distribution shift between offline and online data. Moreover, they usually enforce structural consistency between prior and online policies (e.g., both must be RNNs) (Zhong et al., 2025; Nakamoto et al., 2023). To address these limitations, we introduce prior-guided **cold-start** MARL (see Fig. 1, left), which enables training from scratch using rewarding samples generated by prior policies. Unlike warm-start methods, this paradigm accommodates diverse prior sources, such as rule-based policies derived from large language models (Deng et al., 2025), thereby offering greater flexibility and scalability.

Cold-start effectiveness is evidenced in SARL tasks: for instance, JSRL (Uchendu et al., 2023) leverages imitation-learned prior policies to construct task difficulty based curricula to guide vision-based robotic grasping from scratch (Zhao et al., 2024). However, preliminary experiments in Sec. 4 reveal that although JSRL's curriculum is beneficial in centralized control, naively extending such

Figure 1: Illustration of the proposed DUCE and stylized performance comparison of classical MARL, offline-to-online MARL, and prior-guided cold-start MARL, informed by the results of our preliminary and main experiments. Prior-guided cold-start MARL allows for learning from scratch based on the guidance of prior policies from different sources and of different structures. Under such a paradigm, the proposed DUCE exhibits substantially improved training efficiency.

an idea into MARL actually impairs training efficiency. This is primarily because MARL faces challenges such as environmental non-stationarity and credit assignment, which exacerbate the negative impact of distribution shift introduced by prior policy guidance. On the other hand, in conjunction with the trials presented in Sec. 4 regarding different configurations of policy optimizations, we observe that adding a offline conservative loss can improve early training but leads to collapse. These phenomena are amplified as the number of agents grows, highlighting that the interaction between prior guidance, distribution shift, and value factorization is fundamentally multi-agent rather than a direct carry-over from SARL.

Inspired by William Perry's theory of cognitive development in psychology (Perry Jr, 1999), which emphasizes progressing from teacher-dependent learning to self-directed exploration, we in this paper propose a prior-guided cold-start MARL algorithm named **DUCE**, based on a **du**al-track **c**urriculum learning method for such an **e**xploitation-**e**xploration evolution. Duce coordinates how the prior interacts with the environment and how the resulting data are used for optimization. Its **curriculum based on externally configured task difficulty** is characterized by an interleaved guidance scheme. On starting each episode, a probabilistic decision is made between purely autonomous exploration and curriculum-based guided learning. In the guided case, it gradually reduces the guidance step size (the step length of the prior-controlled prefix of each episode) of the prior policy to construct training tasks that progress from simple to complex. This approach not only provides relatively high-quality samples generated by the prior policy but also ensures a smooth escalation in learning difficulty. The **curriculum based on internally evolved policy optimization** uses a decaying offline loss as a regularization constraint and performs updates multiple times on samples generated by the prior policy, to overcome the exacerbated problem of distribution shift in MARL. While the performance gradually approximates that of the prior policy in the early stages of training, the learning progressively shifts toward online exploration-based policy optimization. Such flexibility allows for consistently enhanced sample efficiency. Overall, these two curricula are designed to co-adapt throughout training in a MARL-specific manner, achieving a synergy between prior-guided exploitation and online environmental exploration that yields both efficient and stable cold-start MARL training.

The key contributions of our work are summarized as follows: 1) We propose DUCE, to the best of our knowledge, as the first prior-guided cold-start MARL method specifically designed to improve training efficiency. Furthermore, it adapts to any architecture of prior policies. 2) We propose a dual-track curriculum learning that synergizes prior guidance with online exploration to accelerate convergence and enhance asymptotic performance. 3) Extensive experiments across various challenging tasks in StarCraft multi-agent challenge (SMAC) v1 and v2 demonstrate that DUCE consistently outperforms state-of-the-art (SOTA) warm-start baselines.

## 2 RELATED WORK

**Offline MARL and O2O MARL.** Value-based MARL, as a pivotal branch, has given rise to a series of seminal algorithms (Qin et al., 2025; 2024; Shen et al., 2023). QMIX (Rashid et al., 2018) is a value-factorization MARL algorithm that tackles the credit-assignment problem by learning per-agent local Q-functions and combining them to compute the global action-value function $Q_{\text{total}}$ through a mixing network with non-negative weights generated by hypernetworks conditioned

on the global state. Its optimized variant, Finetuned-QMIX (ft-QMIX) (Hu et al., 2023), achieves superior benchmark performance (e.g., SMAC) through code-level refinements. Offline MARL instead leverages fixed datasets. For example, CFCQL (Shao et al., 2023) applies counterfactual conservative regularization across agents to compute a global conservative value, while InSPO (Liu et al., 2025) promotes exploration of low-probability actions via sample-wise updates. Adaptive Behavior Cloning (Zhao et al., 2022) adaptively adjusts the BC weight during O2O fine-tuning to avoid sharp performance drops, but relies on a complex PID-style controller and extensive hyper-parameter tuning. In O2O MARL, OVMSE (Zhong et al., 2025) employs a prior Q-network for warm-start initialization and as a basis for constructing temporal-difference targets to guide policy learning. SO2-MADT (Shah et al., 2024) employs transformer architectures suited for long-horizon and multi-agent interactions, but requires high-quality offline data and substantially longer training.

**Prior-Policy-Guided Reinforcement Learning**. This line of research (Uchendu et al., 2023) has primarily focused on single-agent settings, emphasizing how to leverage prior policies derived from large language model and other diverse sources to guide online training. SC-MAIRL (Brackett et al., 2023) targets a semi-centralized execution setting and uses a coarse teacher-guided mechanism with rigid schedules and a reliance on high-quality prior policies. JSRL (Uchendu et al., 2023) is a prior-guided single-agent RL framework combining a guidance policy and an exploration policy; the guidance policy drives the agent to high-value states, while a gradually shrinking guidance horizon reduces prior dependence and facilitates early exploration. PEX (Zhang et al., 2023) and OBAC (Luo et al., 2024) employ Q-values to guide or constrain online training.

**Curriculum learning for RL**. Curriculum learning designs a sequence of tasks of increasing difficulty to improve sample efficiency and stability in RL (Florensa et al., 2017; Narvekar et al., 2020; Ecoffet et al., 2021). Go-Explore (Ecoffet et al., 2021) constructs curricula by assuming the ability to reset to arbitrary states in simulation, whereas JSRL builds curricula solely through prior-policy guidance. Recently, VL-Cogito (Yuan et al., 2025) adopts progressive curricula with adaptive difficulty and dynamic rewards for multimodal reasoning, and RobustDexGrasp (Zhang et al., 2025) combines imitation and reinforcement learning to enhance dexterous grasping robustness. These advances underscore curriculum learning's versatility for addressing complex MARL challenges.

## 3 PRELIMINARY

The MARL problem is formulated within the framework of a Decentralized Partially Observable Markov Decision Process (DecPOMDP) (Bernstein et al., 2013), denoted as $\langle N, \mathcal{S}, \mathcal{A}, P, R, Z, \mathcal{O}, \gamma, H \rangle$. Here, $N$ represents the number of agents, $\mathcal{S}$ the state space with state $s_t$ at time $t$, and $\mathcal{A}$ the joint action space with joint action $\boldsymbol{a}_t$ at time $t$. The environment evolves according to the transition function $P : \mathcal{S} \times \mathcal{A} \times \mathcal{S} \to [0, 1]$, and rewards are given by $R : \mathcal{S} \times \mathcal{A} \to \mathbb{R}$. Since agents have only partial observability, the observation function $Z$ maps states and actions to observations $O_t \in \mathcal{O}$. The process is discounted by factor $\gamma \in (0, 1]$, with a finite horizon $H$. Each agent learns a stochastic policy $\pi : o_t \times a_t \to [0, 1]$ that generates an action–observation history $\tau_t = \{o_0, a_0, \ldots, o_t, a_t\}$. The return is defined as the discounted cumulative reward, $G_t = \sum_{k=0}^{H-t-1} \gamma^k R_{t+k+1}$. The central goal of MARL under this DecPOMDP setting is to optimize policies such that the expected discounted return is maximized, expressed as $J(\pi) = \mathbb{E}\left[\sum_{t=1}^{H} \gamma^t R_t\right] = \mathbb{E}[G_0]$.

## 4 OBSERVATIONS

In this section, we ask whether prior-policy–guided cold-start algorithms developed for single-agent RL can be directly extended to cooperative MARL, and how different policy-optimization mechanisms (purely online vs. offline-regularized) affect training efficiency. We focus on JSRL (Uchendu et al., 2023) as a representative prior-guided single-agent method and study its behavior when combined with standard MARL backbones. Our experiments on SMAC v1 5m_vs_6m-medium and 6h_vs_8z-medium (Fig. 2) reveal two characteristic failure modes, which in turn highlight MARL-specific challenges such as non-stationarity, credit assignment, and the exponentially large joint action space. The additional preliminary results of all aforementioned algorithms, including JSRL-QMIX and JSRL-CFCQL, on 5m_vs_6m-medium-replay and 6h_vs_8z-medium-replay are provided in Appendix A.4.2.

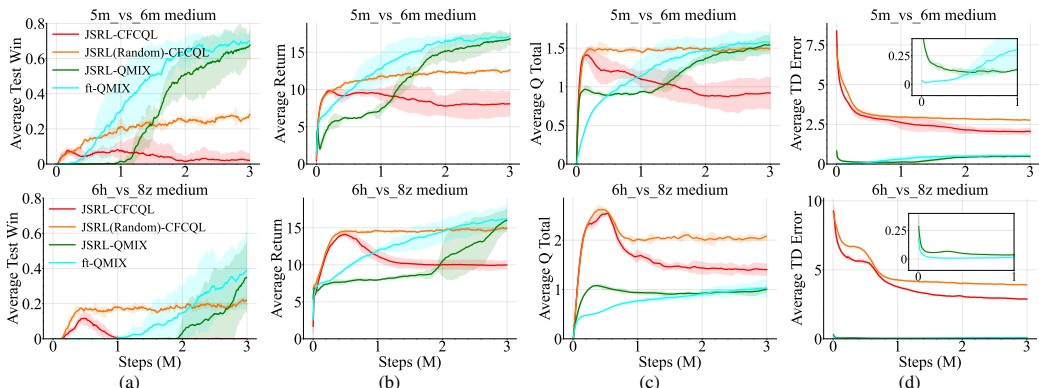

Figure 2: Preliminary results on SMAC v1 5m_vs_6m medium and 6h_vs_8z medium. (a, b) show results of win-rates and returns, whereas (c, d) show results of $Q_{\text{tot}}$ and TD error.

**Observation I. JSRL's curriculum helps in centralized control but degrades ft-QMIX in MARL.** We first examine whether the JSRL curriculum is effective in a centralized single-agent surrogate. To isolate MARL-specific challenges, we formulate the multi-agent task as a centralized control problem in which a single agent observes the global state and outputs the joint action, and we implement both IQL and JSRL-IQL (denotes applying JSRL to an IQL-style single-agent learner.) in this setting (Appendix A.4.3. As shown in Fig. 11, JSRL-IQL achieves a clear early "jump-start" over IQL, confirming that JSRL's curriculum indeed improves training efficiency under centralized control. However, both JSRL-IQL and IQL achieve very low win rates because centralized control still suffers from the curse of dimensionality, which prevents JSRL-IQL from learning effective policies for high-dimensional multi-agent tasks.

We then lift JSRL back to the true multi-agent setting by using ft-QMIX (Hu et al., 2023) as the underlying MARL algorithm. The resulting method, JSRL-QMIX, applies JSRL's prior-guided curriculum on top of a value-factorization backbone and is implemented based on the pymarl3[1] repository (Appendix A.4.1). Fig. 2a and Fig. 2b show that, on 5m_vs_6m-medium and 6h_vs_8z-medium, JSRL-QMIX exhibits lower training efficiency than ft-QMIX during the early and middle stages, despite access to the prior policy. At the same time, Fig. 2c and Fig. 2d indicate that JSRL-QMIX suffers from substantially larger $Q_{\text{tot}}$ and TD error than ft-QMIX in early training. Since JSRL-QMIX differs from ft-QMIX only in how it incorporates prior-policy guidance, this efficiency gap can be attributed to distribution shift between prior-generated and online-generated samples. The enlarged joint action space and non-stationarity in MARL amplify this shift, leading to more severe overestimation/underestimation of Q-values. Moreover, when comparing 5m_vs_6m-medium and 6h_vs_8z-medium, we observe that the instability of JSRL-QMIX becomes more pronounced as the number of agents increases, indicating that the scale of the multi-agent problem has a substantial impact on the effectiveness of prior-guided cold-start methods. Overall, Observation I shows that *although JSRL's curriculum is beneficial in centralized control, a naïve adaptation such as JSRL-QMIX fails to maintain this advantage in MARL and can even harm training efficiency.*

**Observation II. Adding a conservative offline loss improves early training but leads to collapse.** To understand the role of offline conservative regularization, we further extend JSRL-QMIX by incorporating the offline loss from CFCQL (Shao et al., 2023), a representative offline MARL algorithm under CTDE. The resulting method, JSRL-CFCQL, uses a conservative offline loss as a regularizer on top of the prior-guided curriculum. We also consider JSRL(Random)-CFCQL, which replaces JSRL's curriculum with the random-switching scheme from the original JSRL paper—i.e., randomly sampling the number of guide steps at the beginning of each episode.

Fig. 2a and Fig. 2b show that JSRL-CFCQL attains higher win-rates and returns than JSRL-QMIX in the initial phase, confirming that the offline loss can alleviate distribution shift and boost early training by conservatively exploiting prior-policy guidance. However, these gains quickly dissipate: JSRL-CFCQL's performance peaks early and then collapses, eventually performing substantially worse than ft-QMIX. The random-switching variant, JSRL(Random)-CFCQL, avoids the sharp early

---

[1]https://github.com/tjuHaoXiaotian/pymarl3

performance drop and achieves more stable early learning than JSRL-CFCQL, but its progress slows markedly after about 0.5M steps and it still fails to match ft-QMIX in the long run.

From the optimization perspective, these behaviors reflect a trade-off between conservative exploitation and exploration. JSRL-CFCQL uses a strong conservative penalty on prior-guided trajectories, which stabilizes early learning but restricts exploration and biases the replay buffer toward a narrow region of the state space. JSRL(Random)-CFCQL increases sample diversity by randomly varying the guidance length, which mitigates some early over-penalization, yet the lack of an adaptive mechanism to gradually reduce prior dependence leaves later-stage learning constrained by the prior's limited capability. In both cases, the $Q_{tot}$ curves in Fig. 2c and Fig. 2d rise and then fall, stabilizing at relatively high or low levels while maintaining consistently high TD error, indicating more severe overestimation/underestimation than in JSRL-QMIX.

As the number of agents increases (e.g., from 5m_vs_6m-medium to 6h_vs_8z-medium), both JSRL-CFCQL and JSRL(Random)-CFCQL exhibit even stronger instability and collapse in overall performance, $Q_{tot}$, and TD error. This underscores that the scale and interaction complexity of MARL significantly magnify the limitations of prior-guided plus offline-regularized training. Fig. 2b further illustrates that JSRL-CFCQL learns rapidly but collapses after about 0.5M steps, whereas ft-QMIX improves steadily and surpasses it around 1M steps: the former excels at short-term conservative exploitation of prior guidance, while the latter relies on long-term online exploration, yet neither alone is sufficient to solve cold-start MARL from scratch.

Taken together, Observations I and II reveal that (i) JSRL's curriculum is effective in centralized single-agent control but causes severe distribution-shift–induced overestimation when naïvely combined with ft-QMIX in MARL; and (ii) adding a static offline conservative loss can improve early training but leads to collapse without an adaptive mechanism that balances prior-guided exploitation with online exploration. These structural failure modes motivate the dual-track DUCE framework proposed in Sec. 5, which explicitly coordinates task-difficulty scheduling and policy-optimization scheduling in a MARL-specific manner.

## 5 METHOD

In this work, we consider MARL settings subject to maximum step-length constraints, where only suboptimal prior policies without assumptions on their form or structure are available. This choice interprets typical practical situations in which reasonably performing but non-expert controllers are much easier to obtain than near-optimal experts. We define the suboptimality of the prior as achieving performance superior to random policies yet still significantly below the optimal policy. Motivated by the idea of progressing from teacher-dependent learning to self-directed exploration, we propose the DUCE algorithm to synergize prior-guided exploitation and online environmental exploration. Specifically, DUCE establishes a dual-track curriculum learning: one track guided by externally configured task difficulty, and the other driven by internally evolved policy optimization as shown in Fig. 3.

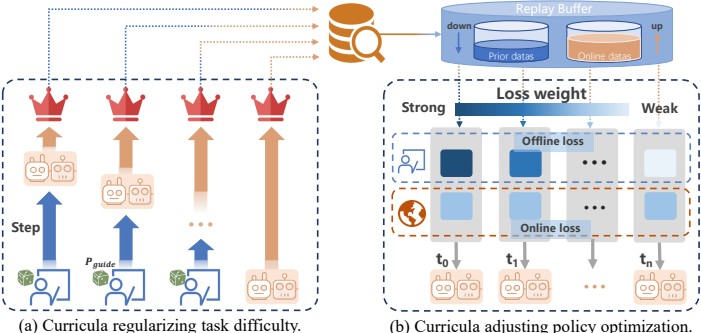

(a) Curricula regularizing task difficulty.    (b) Curricula adjusting policy optimization.

Figure 3: Schematic illustration of the dual-track curriculum setting in DUCE. In (a), the arrow length represents the interaction horizon with the environment; in (b), the upper boxes, whose color fades from dark to light over iteration, reflect the decaying weight of the offline loss, while the lower boxes with constant color represent the fixed weight of the online loss.

**Curriculum based on externally configured task difficulty**. To simultaneously reduce training task difficulty and mitigate the impact of the offline loss on exploration, DUCE adopts an interleaved curriculum built upon JSRL. The only new hyperparameter in this track is the probability $p_{\text{guide}}$, which at the beginning of each episode decides whether the agents perform purely autonomous exploration or follow the prior-guided curriculum; Sec. 6.3 shows that DUCE is robust over a broad range of $p_{\text{guide}}$ values (we use $p_{\text{guide}} = 0.5$ by default). The remaining curriculum parameters are inherited from JSRL: $h$ denotes the number of steps for which the prior policy interacts with the environment (with $h_0$ initialized as the average episode length under the prior alone), $m$ is the number of curriculum stages, and $\beta \in [0, 1]$ is the performance threshold for stage progression. In our experiments, we set $m = 10$ for most tasks (with slightly larger values only for very long-horizon maps) and fix $\beta = 0.95$ across all experiments; sensitivity analyses in Sec. 6.3 indicate that DUCE is empirically robust to moderate changes in these parameters. The curriculum progressively reduces the guidance step size $h$, so that during each stage the mixed policy first executes the prior policy for $h$ steps and then lets the online policy control the remaining $H - h$ steps, effectively shortening the decision horizon in early training and gradually extending it as the curriculum advances.

**Curriculum based on internally evolved policy optimization**. Building on the evolving data distribution induced by the task-difficulty curriculum, the second track dynamically adjusts the policy-optimization mechanism across training. Specifically, we introduce a decaying offline loss as a regularization term on top of the online loss. Since ft-QMIX is a widely adopted MARL algorithm and CFCQL is a classical offline MARL algorithm known for its stability and efficiency, we employ the online loss from ft-QMIX and the offline loss from CFCQL. The online MARL loss combines local values into a joint value $Q_{\text{tot}}$ via a non-negative mixing network to compute TD errors. The offline MARL loss employ a counterfactual strategy: when computing the regularizer for agent $i$, only this agent's action is replaced with an OOD action, while the actions of other agents are fixed to those in the dataset. The loss function is defined as Eq. (1).

$$
\begin{aligned}
\mathcal{L} &= \mathcal{L}_{\text{QMIX}} + \alpha(t)\mathcal{L}_{\text{CFCQL}} \\
&= \left[ Q_{\text{tot}}(s, \boldsymbol{a}) - \left( r + \gamma \max_{\boldsymbol{a}'} \bar{Q}_{\text{tot}}(s', \boldsymbol{a}') \right) \right]^2 + \\
&\quad \alpha(t)\mathbb{E}_{s\sim D}\left[ \sum_{i=1}^{n} \lambda_i \mathbb{E}_{\boldsymbol{a}_{-i}\sim\boldsymbol{\eta}_{-i}} \left[ \log \sum_{a^i} \exp(Q_{\text{tot}}(s, \boldsymbol{a})) \right] - \mathbb{E}_{\boldsymbol{a}\sim\boldsymbol{\eta}}[Q_{\text{tot}}(s, \boldsymbol{a})] \right],
\end{aligned}
\tag{1}
$$

where $\forall i, s, \lambda_i(s) = \exp\left( \tau\mathbb{E}_{\pi_i} \frac{\pi_i(s)}{\eta_i(s)} \right) \Big/ \sum_{j=1}^{n} \exp\left( \tau\mathbb{E}_{\pi_j} \frac{\pi_j(s)}{\eta_j(s)} \right)$, $\eta$ denotes the behavior policy that generates the offline dataset used in $\mathcal{L}_{\text{CFCQL}}$, and it is distinct from the prior policies used in the external task-based curriculum; this decoupling allows DUCE to remain compatible with prior policies of different architectures. $s'$ and $\boldsymbol{a}'$ denote the subsequent state and joint action, respectively. $\boldsymbol{a}_{-i}$ denotes the joint action of all agents except agent $i$. $\bar{Q}_{\text{tot}}$ denotes the target joint action-value function. The computation of $\pi(s)$ and $\eta(s)$ follows the formulation in CFCQL; for the detailed derivation, please refer to Appendix B.2 of that paper (Shao et al., 2023). $\alpha(t)$ represents the weighting factor of the offline loss $\mathcal{L}_{\text{CFCQL}}$, which is a linear decay function with time $t$. The value of $\alpha(t)$ decreases linearly from $\alpha_{\text{start}}$ to 0 within $T_\alpha$ steps, defined as $\alpha(t) = \max\left( 0, \alpha_{\text{start}} - \frac{\alpha_{\text{start}}}{T_\alpha} \cdot t \right)$. Among the hyperparameters in this track, the decay period $T_\alpha$ is the only new one introduced by DUCE. $\alpha_{\text{start}}$ is inherited from CFCQL. Empirically, DUCE is robust to moderate changes in these two hyperparameters, as illustrated by the sensitivity analyses reported in Sec. 6.4.

Following the optimization-based curriculum, in the early training phase, the offline loss carries large weight, thereby accelerating training efficiency. As the weight of the offline loss gradually decays and the policy update frequency approaches normal, the policy optimization progressively shifts toward online environmental exploration. Overall, the policy-optimization curriculum defines a progression of optimization regimes that exploits complementary advantages by gradually shifting updates from conservative to exploratory; prior-dominated data is handled conservatively, while later online data is allowed to drive exploration.

DUCE's core novelty lies in the tight integration and co-adaptation of its two curricula throughout training. The task-difficulty curriculum drives a mixed prior–online policy to collect higher-quality samples, populating the replay buffer with prior-guided data. The policy-optimization curriculum then performs conservative updates on this mixed distribution to accelerate early learning despite

distribution shift. As the online policy improves and the task-difficulty curriculum advances, autonomous exploration increases, shifting the replay buffer toward on-policy data. The optimization curriculum correspondingly transitions from conservative to exploration-driven updates. Thus, the two curricula operate synergistically: early learning leverages prior-guided samples with conservative optimization, while later learning relies on growing exploration data with progressively more exploratory updates. Additionally, in Appendix A.6, we provide theoretical analysis of DUCE, showing that its dual-track curriculum that we propose provably attains polynomial sample complexity.

## 6 EXPERIMENTS

We evaluate whether DUCE improves MARL training efficiency in cold-start settings by leveraging suboptimal prior policies. Specifically, we investigate the following research questions: (1) Can DUCE generally enable efficient MARL training? (2) How critical is the task-based curriculum for DUCE? (3) How critical is the optimization-based curriculum for DUCE? (4) Can DUCE utilize diverse sources of prior policies?

### 6.1 BASELINES AND EXPERIMENTAL SETUP

**Baselines**. Given the absence of existing studies on sophisticated prior-policy-guided and cold-start MARL, we benchmark DUCE against five warm-start SOTA methods: (1) OVMSE (Zhong et al., 2025), the current SOTA O2O MARL approach without an official implementation, for which we reimplemented Offline Value Function Memory (OVM) and Sequential Exploration (SE) on the repositories of pymarl3 and CFCQL following the original paper; (2) InSPO[2] (Liu et al., 2025), a leading offline MARL algorithm; (3) CFCQL[3] (Shao et al., 2023), a canonical offline MARL method; (4) MACal-QL[4] (Nakamoto et al., 2023), which applies the classic single-agent O2O algorithm Cal-QL directly to MARL; and (5) ft-QMIX (Hu et al., 2023), a classical MARL method. In line with recent O2O designs that maintain loss-function consistency, InSPO, CFCQL, and MACal-QL use the same losses in both offline and online phases. OVMSE adopts the CFCQL loss offline, while—following the emphasis of the original paper on avoiding offline-RL losses online—it uses the ft-QMIX loss during the online phase. To strengthen these warm-start baselines, their online training is allowed to leverage offline data, where at each gradient step, 30% of the training samples are drawn from the offline buffer and 70% from online interaction. This design choice follows the empirical finding in prior work (Zhou et al., 2025) that training solely on online experience during fine-tuning, without retaining offline data, can severely degrade how well the model fits the offline distribution. To verify this phenomenon in our setting, Appendix A.5.2 presents an ablation in which offline data is entirely removed during online training, resulting in a deterioration in performance. For fairness, DUCE employs the same prior policy as that produced by CFCQL and OVMSE in the offline stage. In this work, DUCE adopts ft-QMIX as the underlying multi-agent reinforcement learning algorithm, whose implementation is based on the repositories of pymarl3[5] and CFCQL[3]. Further details appear in Appendix A.3.

**Experimental setup**. We conduct experiments on widely recognized SMAC v1 and v2 benchmarks, including four challenging SMAC v1 tasks: 5m_vs_6m (hard), 6h_vs_8z (super hard), Corridor (super hard), and MMM2 (super hard), as well as two representative SMAC v2 tasks: Protoss 5v5 and Zerg 5v5. Since existing O2O algorithms require pretraining on offline datasets, we use the same dataset for all methods on each task to ensure fair comparison. Each task is associated with two dataset variants, i.e., medium and medium-replay. For 5m_vs_6m and 6h_vs_8z, we directly adopt the offline datasets provided by CFCQL. For the other tasks, datasets are constructed following the procedure described in the CFCQL paper: the medium dataset is collected by executing an ft-QMIX policy with moderate performance, while the medium-replay dataset is taken from the replay buffer during ft-QMIX training when the policy reaches moderate performance. The medium datasets contains 5,000 trajectories, while the medium-replay datasets contain no more than 5,000 trajectories. Further dataset details appear in Appendix A.2. DUCE and the other baselines' hyperparameter

---

[2]https://github.com/kkkaiaiai/InSPO

[3] https://github.com/thu-rllab/CFCQL

[4]https://nakamotoo.github.io/Cal-QL

[5]https://github.com/tjuHaoXiaotian/pymarl3

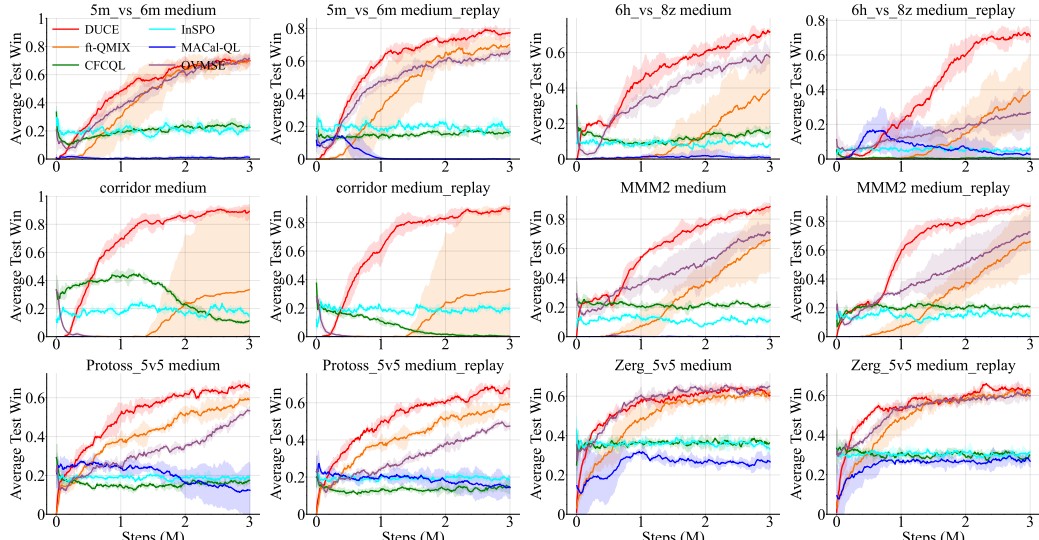

Figure 4: Comparison of DUCE with ft-QMIX and other MARL baselines across various maps.

settings are provided in Appendix A.3.1 and A.3.1. Moreover, Appendix A.5.3 reports experiments evaluating DUCE on the widely recognized Multi-Agent MuJoCo benchmark, demonstrating its applicability to continuous-action multi-agent settings.

## 6.2 CAN DUCE ENABLE EFFICIENT MARL TRAINING?

As shown in Fig. 4, DUCE markedly outperforms ft-QMIX and the other four SOTA baselines, converging faster to a higher asymptotic performance, even though the other four baselines employ warm starts that confer an initial advantage. By 1M steps, DUCE surpasses all four warm-start baselines on most tasks with a 10-40% win-rate lead. By 2M steps, this grows to 10-60% on most maps. At 3M steps, DUCE gradually eventually stabilizes, maintaining a 10-50% win-rate advantage across most maps while continuing to improve. Notably, it achieves around 80% win-rates even on super hard settings of 6h_vs_8z, corridor and MMM2, far exceeding competitors. These results indicate that DUCE can enable efficient MARL training.

InSPO and CFCQL maintain around 20% win-rates on most tasks, mainly because offline losses penalize transitions involving unseen state–action pairs, thus suppressing exploration, leading to repeated collection of near-duplicate samples, and ultimately causing overfitting. For the same reasons, MACal-QL performs even worse. OVMSE surpasses ft-QMIX on only half of the tasks; its win-rate and return curves show a pronounced initial drop and slow recovery. This is primarily because OVMSE relies on prior $Q_{tot}$ estimates to build TD targets, but these are unreliable for out-of-distribution states, leading policy updates astray. Although both ft-QMIX and DUCE are trained from scratch, DUCE merely incorporates samples generated by a prior policy at the start of training, yet achieves a significant advantage in sample efficiency. This is largely attributed to the dual-track curriculum learning design of DUCE, which not only structures training tasks in a progressive manner, but also dynamically adjusts the policy optimization mechanism. This enables a synergy between prior-guided exploitation and online environmental exploration. To more clearly analyze the improvements introduced by DUCE, Appendix A.5.10 presents a qualitative visualization of the state distributions visited by the prior and online policies.

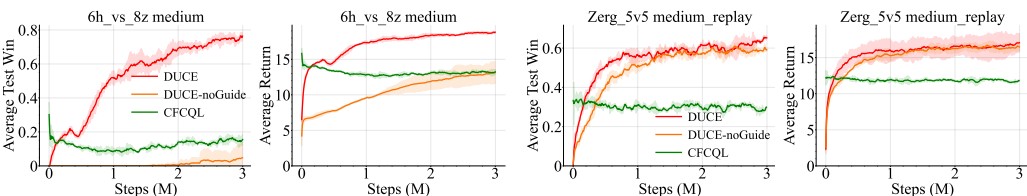

Figure 5: Ablation study of the task-based curriculum.

### 6.3 HOW CRITICAL IS THE TASK-BASED CURRICULUM FOR DUCE?

As shown in Fig. 5, DUCE-noGuide refers to the variant of DUCE without the task-based curriculum. On the 6h_vs_8z-medium task, DUCE-noGuide performs substantially worse than DUCE, achieving a win-rate below 10%. On the Zerg_5v5-medium_replay task, compared with DUCE, it shows lower training efficiency during the early stages. These results demonstrate that the curriculum of externally configured task difficulty plays a critical role in enhancing the training efficiency of MARL, particularly during the initial phase. In addition, Appendix A.5.9 includes ablations on per-agent curricula, showing that making the guidance schedule heterogeneous across agents increases training variance without providing consistent gains in CTDE value-factorization setting.

As shown in Fig. 6a, we varied the interleaving probability $p_{\text{guide}}$ across 0.1, 0.3, 0.5, 0.7, and 1. When $p_{\text{guide}}$ is below 0.5, the prior-policy guidance becomes insufficient, leading to a significant decline in overall training efficiency. When $p_{\text{guide}} = 1$, DUCE improves quickly at the start but then experiences a sharp drop and large oscillations. In contrast, $p_{\text{guide}} < 1$ yields slower early gains but produces more stable learning curves and slightly better mid- to late-stage performance. Overall, when $0.5 \leq p_{\text{guide}} < 0.7$, its incorporation helps alleviate the restrictive influence of offline loss on exploration, thereby preventing substantial performance fluctuations. To further illustrate the effect of introducing $p_{\text{guide}}$, Appendix A.5.7 reports additional ablation experiments on Corridor and 5m_vs_6m under both medium and medium-replay settings. As shown in Fig. 6b, we varied the performance threshold $\beta$ with values of 1, 0.95, 0.9, 0.85, and 0.8, and observe slight fluctuations in the early training phase when $\beta < 0.9$. Overall, DUCE demonstrates strong robustness to the choice of $\beta$.

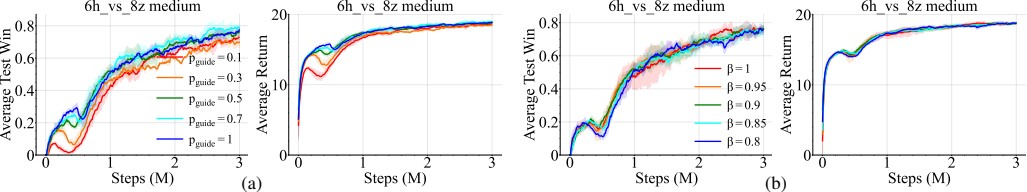

Figure 6: Sensitivity analysis of interleaving probability $p_{\text{guide}}$ (a) and performance threshold $\beta$ (b).

### 6.4 HOW CRITICAL IS THE OPTIMIZATION-BASED CURRICULUM FOR DUCE?

In Fig. 7, DUCE-noOnlineLoss denotes the removal of $\mathcal{L}_{\text{QMIX}}$ from the DUCE loss function while keeping $\alpha(t)$ fixed as a time-invariant constant, wheras DUCE-noOfflineLoss refers to eliminating $\mathcal{L}_{\text{CFCQL}}$. By removing the optimization-based curriculum, these two variants represent two cases of policy optimization mechanisms: one driven solely by offline loss and the other solely by online loss. DUCE-noOnlineLoss leads to training divergence (see Fig. 7a) and overfitting (see Fig. 7b), primarily due to the offline loss imposing excessive penalties on unseen state–action transitions and persistently underestimating global state–action values. DUCE-noOfflineLoss performs worse than DUCE on the 6h_vs_8z-medium task, mainly because it struggles to address distributional shifts induced by prior policy guidance. On Zerg_5v5-medium_replay, the performance degradation of DUCE-noOfflineLoss is negligible compared with DUCE, as the foundational MARL algorithm, ft-QMIX, inherently exhibits high training efficiency in the early stages of this task, rapidly reducing the distribution mismatch induced by prior policy guidance. In summary, the curriculum of internally evolved policy optimization is crucial for leveraging prior policy guidance to mitigate the challenges of training MARL agents from scratch. Moreover, Appendix A.5.8 presents a systematic evaluation of different decay schedules in the optimization-based curriculum.

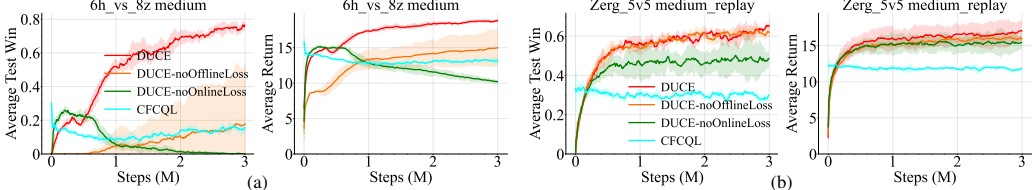

Figure 7: Ablation study of the optimization-based curriculum.

In Fig. 8a, we examine the effect of varying the initial weight $\alpha_{\text{start}}$ of the decaying offline loss in the optimization-based curriculum, setting it to 1, 3, 5, 7, and 10. We observe that a small $\alpha_{\text{start}}$ leads to inferior performance before 0.5M steps, while an excessively large value causes considerable training fluctuations in the early phase. Nevertheless, after 0.5M steps, the performance under all the configurations improves rapidly. Overall, the choice of $\alpha_{\text{start}}$ exerts limited influence on the overall training efficiency. In Fig. 8b, we evaluate the decay period $T_\alpha$ of the decaying offline loss in the optimization-based curriculum with values of 0.1M, 0.3M, 0.5M, 0.7M, and 1M. The results show that when $T_\alpha$ is smaller than 0.5M or larger than 0.7M, the training efficiency in the early stage is substantially reduced, while the mid- and late-stage performance remains unaffected. Overall, as long as $T_\alpha$ is not excessively large, its impact on the overall training efficiency is minimal.

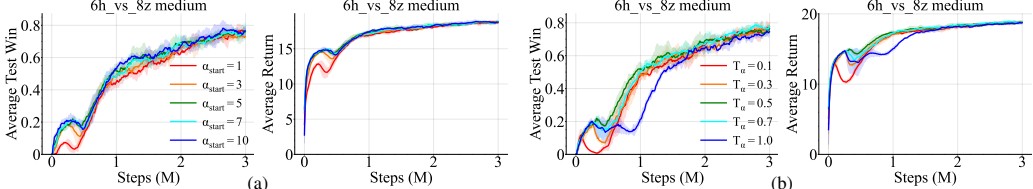

Figure 8: Sensitivity analysis of the decaying offline loss in the optimization-based curriculum: (a) varying initial weight $\alpha_{\text{start}}$; (b) varying decay period $T_\alpha$.

### 6.5 CAN DUCE UTILIZE DIVERSE SOURCES OF PRIOR POLICIES?

As shown in Fig. 9, DUCE_FC_Model denotes the variant that replaces the recurrent neural network (RNN)-based prior policy, originally obtained via CFCQL, with a fully connected network–based prior policy for guidance. DUCE_LLM_Rule refers to the use of rule scripts generated by a large language model as the guiding prior policy. Given the high difficulty of the 6h_vs_8z and Zerg_5v5 tasks, even with the aid of a large model, the generated rule scripts achieve a win-rate of zero and provide only limited heuristic guidance, such as focus-fire maneuvers. Nevertheless, both DUCE_FC_Model and DUCE_LLM_Rule significantly accelerate training compared to ft-QMIX. These results indicate that DUCE can effectively exploit prior policies of varying origins, forms, and structures to improve the efficiency of MARL training from scratch. Moreover, DUCE is not strongly dependent on the quality of the prior and can also incorporate stronger (expert-level) priors. Appendix A.5.4 reports experiments with expert priors that empirically verify this property.

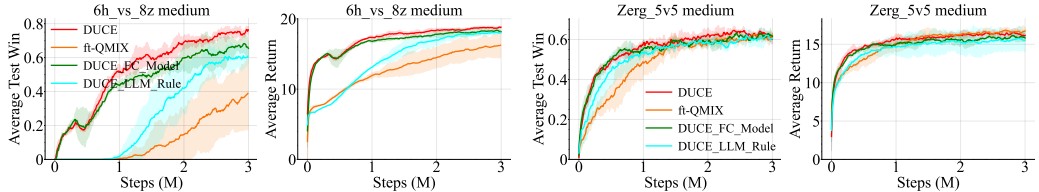

Figure 9: Training performance of DUCE using different prior policy types, including fully connected (FC) models and LLM-generated rule scripts.

## 7 CONCLUSION

To leverage prior policies from diverse sources to enhance the efficiency of MARL training from scratch, we introduce prior-guided cold-start MARL. Such a setting is particularly important for extending the applicability of MARL to real-world tasks, offering greater flexibility and scalability. However, naive prior-guided cold-start methods adapted from SARL fail to mitigate distribution shift or adapt policy optimization across training phases, thus reducing efficiency. To this end, we propose the DUCE approach, which designs a dual-track curriculum learning, with one track guided by the externally configured task difficulty and the other driven by the internally evolved policy optimization. These two tracks operate in concert for a synergy between prior-guided exploitation and online environmental exploration to ensure both efficiency and stability during the cold-start MARL training phase. On the widely used SMAC v1 and v2 benchmarks, DUCE consistently outperforms SOTA warm-start baselines.

## ETHICAL STATEMENT

This study constitutes algorithmic research carried out in a simulation environment, involving no experiments on human or animal subjects, nor utilizing any datasets containing personal privacy or sensitive information. All experiments were carried out in the open multi-agent benchmark environment SMAC v1 and v2.

Multi-agent systems and their decision-making algorithms hold broad potential for societal applications, such as autonomous driving and robotic collaboration. This research focuses on improving the learning efficiency of agents during the cold-start phase. However, we emphasize that the study itself does not directly involve high-risk applications. prior policies used originate from pre-trained models or human-designed rules within the same simulation environment, aimed at accelerating learning rather than introducing biases. Furthermore, any system based on such algorithms must undergo rigorous validation before practical deployment to ensure the fairness, reliability, and safety of its decisions, thereby avoiding discriminatory or harmful outcomes.

## REPRODUCIBILITY STATEMENT

To fully ensure the reproducibility of the proposed multi-agent reinforcement learning (MARL) cold-start training method based on prior policies, Our code has been submitted anonymously[6].

The appendix of this paper provides complete details necessary for reproducing the experimental results, including key steps for setting up the experimental environment, specific operational instructions, and exhaustive hyperparameter configurations for each experiment (e.g., learning rate, batch size, number of training iterations). This ensures that other researchers can replicate this study under identical conditions. We sincerely encourage academic peers to utilize these resources in broader multi-agent learning scenarios to further validate and extend the findings of this research.

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

# A APPENDIX

## A.1 LLMs USAGE STATEMENT

In the preparation of this manuscript, large language models were solely utilized for auxiliary text polishing purposes. The models did not participate in any core research activities, including but not limited to the conception of research ideas, method design, data analysis, result interpretation, and conclusion derivation. All academic viewpoints, technical details, and scientific assertions presented in this paper are original achievements independently completed by the authors.

We assume full responsibility for the accuracy, completeness, and scientific validity of the entire content of the manuscript. We confirm that the content modified by large language models is free from plagiarism, factual distortion, or any form of academic misconduct.

## A.2 OFFLINE DATASETS

To ensure the validity, reproducibility, and comparability of the experiments, this paper constructs a diverse set of datasets for the offline training scenario in Multi-Agent Reinforcement Learning (MARL), covering different task complexities, agent scales, and action space dimensions. The core information of all datasets (including trajectory statistics, spatial dimensions, and performance metrics) is summarized in Table 1. Detailed explanations are provided below from three aspects: data source and construction logic, dataset classification and definition, and key characteristics and statistical notes.

### A.2.1 DATA SOURCE AND CONSTRUCTION NORMS

The dataset construction strictly adheres to the principle of "benchmark reuse + unified extension" to ensure the consistency of data distribution and the fairness of experimental comparison. The specific details are as follows:

- **Benchmark Dataset Reuse**: The datasets for two classic multi-agent adversarial scenarios, 5m_vs_6m and 6h_vs_8z, are directly sourced from the open-source project of the CFCQL algorithm[7]. This project is a representative work in the field of offline training in MARL, and its datasets have been validated in the field, serving as the benchmark comparison data for the experiments in this paper.

- **Extended Dataset Construction**: The datasets for four scenarios (corridor, MMM2, Protoss_5v5, and Zerg_5v5) are independently constructed in accordance with the data collection specifications of CFCQL. Specifically, trajectories are generated through the buffer during training or the fixed policy after training completion, ultimately forming offline datasets with a unified structure to ensure comparability with the benchmark datasets.

Table 1: Overview of datasets used in experiments, including details of trajectories, samples, agent counts, and state, observation, and action space dimensions, with average returns indicating performance levels.

| Instances | | Trajectories | Samples | Agents | State dim | Obs dim | Action dim | Average returns |
|---|---|---|---|---|---|---|---|---|
| 5m_vs_6m | medium | 5.0K | 135.2K | 5 | 98 | 55 | 12 | 12.05 |
| | medium_replay | 5.0K | 118.4K | 5 | 98 | 55 | 12 | 9.02 |
| 6h_vs_8z | medium | 5.0K | 207.0K | 6 | 140 | 78 | 14 | 16.63 |
| | medium_replay | 5.0K | 182.4K | 6 | 140 | 78 | 14 | 12.97 |
| corridor | medium | 5.0K | 506.3K | 6 | 282 | 156 | 30 | 16.50 |
| | medium_replay | 5.0K | 412.5K | 6 | 282 | 156 | 30 | 13.24 |
| MMM2 | medium | 5.0K | 279.4K | 10 | 322 | 176 | 18 | 11.30 |
| | medium_replay | 5.0K | 243.4K | 10 | 322 | 176 | 18 | 13.68 |
| Protoss_5v5 | medium | 5.0K | 279.5K | 5 | 130 | 92 | 11 | 14.07 |
| | medium_replay | 4.5K | 255.4K | 5 | 130 | 92 | 11 | 11.20 |
| Zerg_5v5 | medium | 5.0K | 162.3K | 5 | 120 | 82 | 11 | 12.87 |
| | medium_replay | 5.0K | 156.1K | 5 | 120 | 82 | 11 | 11.30 |

### A.2.2 DATASET CLASSIFICATION AND DEFINITION

In line with general practices, two types of datasets are designed for each scenario. The core difference between the two types lies in the performance level of the policy used for trajectory sampling, with specific definitions as follows:

- **Medium**: Generated by sampling from a medium-performance policy. This policy must meet the following requirements: after a certain number of training episodes in the target scenario, its performance stably lies between that of a random policy and an optimal policy, and the exploration and exploitation of trajectories reach a balance—avoiding both the ineffective exploration of random policies and the overfitted trajectories of optimal policies,

---

[7]https://github.com/thu-rllab/CFCQL

which is more consistent with the characteristics of non-optimal but effective datasets in real offline scenarios.

- **Medium_replay**: Derived from the Replay Buffer during policy training. Specifically, starting from policy initialization, trajectories are continuously collected and stored in the buffer until the policy performance meets the medium-performance standard, at which point all trajectories stored in the buffer constitute this dataset. Its core characteristic is that it contains data covering the complete learning process of the policy from low performance to medium performance, with a higher diversity of trajectory distribution than the medium dataset.

### A.2.3  SPECIAL NOTES

Except for special annotations, all datasets comply with a unified scale standard: containing 5.0K trajectories. This ensures the consistency of data volume across different scenarios and avoids interference from data volume differences on offline training results. The length of each trajectory (i.e., the number of samples) varies with scenario complexity: simple adversarial scenarios (5m_vs_6m, 6h_vs_8z) have shorter trajectories, while the corridor scenario requires agents to complete long-range path planning and collaboration, resulting in the highest number of samples among all scenarios.

The medium_replay dataset for the Protoss_5v5 scenario is the only exception, containing only 4.5K trajectories (failing to meet the unified standard of 5.0K). The reason is that during the policy training process for this scenario, when the number of trajectories accumulated in the replay buffer reached 4.5K, the policy performance had already met the judgment threshold for medium performance in advance.

### A.3  IMPLEMENTATION DETAILS

Detailed implementation specifics of each baseline algorithm have been fully documented in Appendix A.3.1 for reference.

The experimental code of this paper is developed based on the pymarl3 framework[8], where the implementation of the offline training module refers to the core implementation of CFCQL[7]. To lower the threshold for experiment reproduction, the code enhances tolerance toward dependencies of the development environment and version compatibility. Meanwhile, it is accompanied by standardized configuration scripts and one-click running scripts, which can support rapid deployment and result reproduction. In terms of hardware configuration, the offline loss calculation process in some experimental environments has high requirements for video memory resources. Therefore, the experiments in this study are mainly conducted on graphics cards of GeForce RTX 3090 (24GB), GeForce RTX 4090 (24GB), NVIDIA A100-SXM4 (40GB) or GeForce RTX 4090 PLUS (48GB) to ensure computational stability and efficiency.

To guarantee the reliability and statistical significance of the experimental results, all experiments in this paper are repeated and verified using 3 independent random seeds; during each model evaluation, 16 consecutive test rounds are conducted, and the average value of the results is taken as the win-rate and reward metrics of the current strategy. Additionally, the exponential moving average method is adopted for smoothing when plotting the experimental data charts, with the smoothing balance coefficient set to 0.9. This measure effectively reduces the interference caused by data fluctuations and more clearly presents the variation trend of the strategy performance.

### A.3.1  BASELINE ALGORITHM

**1) ft-QMIX (Hu et al., 2023)**  We directly use the pymarl3 implementation of ft-QMIX, including all hyperparameters for the various SMAC v1 and v2 maps, strictly following recommended values of hyperparameters provided by the repository. For DUCE, CFCQL, MACal-QL and OVMSE that build on ft-QMIX, all hyperparameters such as buffer size, learning rate, batch size, and optimizer, follow the ft-QMIX defaults unless explicitly modified by the corresponding algorithm.

---

[8]https://github.com/tjuHaoXiaotian/pymarl3

**2) CFCQL (Shao et al., 2023)** CFCQL[7] remains largely consistent with the original source code. Initially, offline data must be prepared and written into an offline buffer. Prior to the initiation of training, the behavioral model is trained using the offline buffer, followed by the execution of offline training. Upon completion of the offline phase, the process transitions to online training, where a new empty online buffer is established to store trajectories sampled online by the exploration strategy. Samples from both the offline buffer and the online buffer are aggregated to form the final sample set for a single training iteration, which is subsequently employed for loss computation and parameter updates.

**3) InSPO (Liu et al., 2025)** For InSPO, we have directly integrated online sampling and an online buffer module into its source code[9], with the design aligned with that of CFCQL.

**4) MACal-QL (Nakamoto et al., 2023)** The implementation of MACal-QL is straightforward. We referenced the source code of Cal-QL[10] and implemented it based on the MACQL method in CFCQL. First, we calculate the Monte Carlo return of the behavioral strategy as the value of the reference strategy, and then take the maximum value between the Q-value of the action sampled by the strategy and the value of the reference strategy. The offline-to-online design is consistent with that of CFCQL.

**5) OVMSE (Zhong et al., 2025)** The source code of OVMSE is not publicly available. Therefore, we reconstructed the two modules, namely OVM and SE, by referring to the formulas and pseudocode presented in the original paper. Specifically, for the OVM module, it is only necessary to retain the offline pre-trained target value function; when calculating the target Q-value, the maximum value between the offline memory value and the online TD target value is adopted. In contrast, the implementation of the SE module is more straightforward, which merely involves reducing the exploration probability in proportion to the number of agents. Additionally, the transition process from the offline phase to the online phase in our reconstruction aligns with the design principle of the original OVMSE. In the experiments, the performance of our implementation of OVMSE failed to match the level reported in the original paper on certain datasets. This discrepancy is not intentionally induced to degrade its performance; instead, it stems from the limited efficacy of the OVM mechanism in these specific map environments. To address this issue, we have specifically shortened the decay cycle of the OVM (it should be noted that OVMSE does not provide open access to its detailed parameter configurations). However, the OVMSE mechanism, due to its inherent simplicity, exhibits insufficient effectiveness in tasks (such as the task on Corridor) that impose high requirements on exploration. This limitation results in an almost zero win-rate for OVMSE on the corridor map.

### A.3.2 HYPER-PARAMETERS

DUCE's dual-track design introduces only two new hyperparameters: the interleaving probability $p_{\text{guide}}$ and the decay period $T_\alpha$ of the offline loss. Conceptually, $p_{\text{guide}}$ controls how often the agents follow prior-guided episodes versus purely online exploration, while $T_\alpha$ controls the decay period of conservative offline loss during training. Our ablations in Sec. 6.3 and 6.4 show that, as long as these two hyperparameters stay within a reasonably wide operating range around their default values, the performance curves change only mildly; noticeable degradation occurs only when they are pushed to extreme values (e.g., $p_{\text{guide}}$ very close to 0, or $T_\alpha$ extremely large or extremely small). We recommended $p_{\text{guide}} = 0.5$ and $T_\alpha = 100K$ as default values, allowing readers to adopt them directly without further tuning.

We also inherit several parameters from prior work: the number of curriculum stages $m$ and the performance threshold $\beta$ from JSRL, and the initial offline-regularization weight $a_{\text{start}}$ from CFCQL. These hyperparameters are all easy to tune. The JSRL paper notes that $m$ only needs to be large enough to allow smooth curriculum progression; in our experiments, we set $m = 10$ for most tasks. Both $\beta$ and $a_{\text{start}}$ have clear recommended ranges in their respective papers. We adopt $\beta = 0.95$ and $a_{\text{start}} = 1$ as default values, and the sensitivity analyses in Sections 6.3 and 6.4 show that moderate

---

[9]https://github.com/kkkaiaiai/InSPO
[10]https://nakamotoo.github.io/Cal-QL

variations of these parameters lead to only minor performance differences. Overall, DUCE does not introduce a heavy hyperparameter-tuning burden.

Table 2: DUCE hyperparameters across different maps.

| Map | Guide ($p_{\text{guide}}/\beta/m$) | CFCQL $\alpha$ ($\alpha_{\text{start}}/T_{\alpha}$) |
|---|---|---|
| 5m_vs_6m (medium) | $(0.5/0.95/10)$ | $(3/100k)$ |
| 5m_vs_6m (medium_replay) | $(0.5/0.95/10)$ | $(3/100k)$ |
| 6h_vs_8z (medium) | $(0.5/0.95/10)$ | $(5/500k)$ |
| 6h_vs_8z (medium_replay) | $(0.5/0.95/10)$ | $(3/500k)$ |
| MMM2 (medium) | $(0.5/0.95/10)$ | $(10/500k)$ |
| MMM2 (medium_replay) | $(0.5/0.95/10)$ | $(10/500k)$ |
| corridor (medium) | $(0.5/0.95/20)$ | $(1/20k)$ |
| corridor (medium_replay) | $(0.5/0.95/20)$ | $(1/20k)$ |
| Protoss_5v5 (medium) | $(0.5/0.95/10)$ | $(1/100k)$ |
| Protoss_5v5 (medium_replay) | $(0.5/0.95/10)$ | $(1/100k)$ |
| Zerg_5v5 (medium) | $(0.5/0.95/10)$ | $(1/100k)$ |
| Zerg_5v5 (medium_replay) | $(0.5/0.95/10)$ | $(1/100k)$ |

## A.4 PRELIMINARY EXPERIMENTS

This section primarily supplements Sec. 4, including implementation details and additional preliminary experiments.

### A.4.1 IMPLEMENTATION DETAILS OF PRELIMINARY EXPERIMENTS

In our implementation, JSRL is built upon the pymarl3 framework. The experimental hyperparameters are aligned with those in DUCE. In the preliminary experiments, the prior policies are identical to those used in the main results of Sec. 6.2, with detailed descriptions provided in Table 4. For instance, on the 6h_vs_8z medium-replay task, the prior policy is obtained by training CFCQL on the corresponding medium-replay offline dataset, achieving an average win rate of 5% and an average return of 13.33.

### A.4.2 PRELIMINARY EXPERIMENTS RESULTS ON MEDIUM-REPLAY TASKS

As shown in Fig. 10, these are the preliminary results of ft-QMIX, JSRL-QMIX, JSRL-CFCQL, and JSRL(Random)-CFCQL on 5m_vs_6m-medium-replay and 6h_vs_8z-medium-replay.

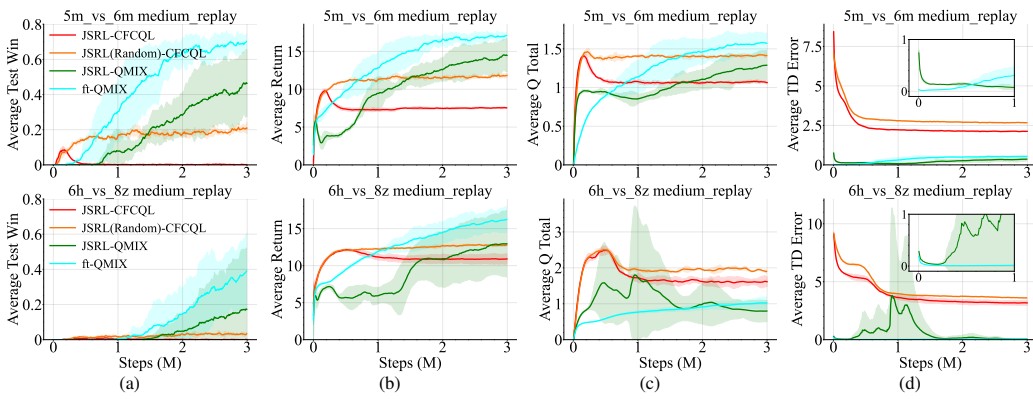

Figure 10: Preliminary results on SMAC v1 5m_vs_6m medium_replay and 6h_vs_8z medium_replay. (a, b) show results of win-rates and returns, whereas (c, d) show results of $Q_{\text{tot}}$ and TD error.

### A.4.3 COMPARISON EXPERIMENTS BETWEEN IQL AND JSRL-IQL

To isolate the effect of MARL-specific challenges (non-stationarity and credit assignment), we conduct a centralized single-agent experiment. Specifically, we formulate a multi agent problem as a centralized control problem, where a single agent observes the global state and outputs the joint action. On top of this centralized formulation, we implement IQL and JSRL-IQL, i.e., applying JSRL to an IQL-style single-agent learner.

As shown in Fig. 11, the reward curves clearly indicate in the early stage of training, JSRL-IQL exhibits a rapid performance increase compared with IQL, demonstrating that in the centralized setting, JSRL-IQL indeed provides a clear "jump-start" over the vanilla IQL baseline. This confirms that the JSRL curriculum is beneficial relative to no curriculum in the centralized formulation, which is precisely why we adopted JSRL as a promising starting point. However, centralized control suffers from the curse of dimensionality, making it difficult for JSRL-IQL to learn effective policies. This indicates that the core challenge does not lie in JSRL itself, but in extending prior-guided cold-start training to factorized MARL under Dec-POMDPs, which is precisely what DUCE is designed to address.

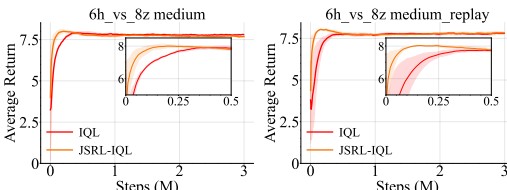

Figure 11: Performance comparison between IQL with and without guidance.

### A.4.4 COMPARISONS BETWEEN THE PRIOR POLICY AND THE ONLINE POLICY DURING TRAINING

In this section, we conduct an experiment illustrating the performance of the guidance policy, the exploration policy, and the combined policy in JSRL-CFCQL during preliminary runs on 6h_vs_8z-medium and 6h_vs_8z-medium-replay. The guidance policy refers to the prior policy, which is frozen during online training and plays the role of a teacher guiding the learning process of the exploration policy. For example, on the 6h_vs_8z medium-replay task, the prior policy is obtained by training CFCQL on the medium-replay offline dataset, achieving an average win rate of 5% and an average return of 13.33. The performance of prior policies for other tasks is reported in Table 4. The exploration policy refers to the MARL policy that is learned online through trial and error. The combined policy denotes the policy formed by composing the guidance and exploration policies: the guidance policy interacts with the environment for a prefix of each episode, after which the exploration policy continues until termination.

In Fig. 12, the return curves clearly show that the guidance policy maintains constant performance throughout training and substantially outperforms the exploration policy at initialization; in the early phase, the mixed policy remains close to the guidance policy, while the exploration policy gradually improves. When the curriculum completes and the guidance policy no longer interacts with the environment, the combined policy reduces to the exploration policy alone. After the curriculum, however, the conservative regularization from CFCQL imposes excessive penalties that constrain exploration, restrict the collected samples to a narrow local region of the state space, inflate in-distribution Q-values, and ultimately cause overfitting—highlighting that the core challenge in MARL lies in how prior guidance is incorporated.

### A.5 ADDITIONAL EXPERIMENTAL

### A.5.1 BASELINE PERFORMANCE SUPPLEMENT

Fig. 13 compares the performance of DUCE with other baseline algorithms in terms of average test return. Across nearly all maps, DUCE achieves convergence by 1M training steps. Notably, DUCE consistently outperforms the other algorithms, with the advantage becoming especially pronounced in the most challenging scenarios.

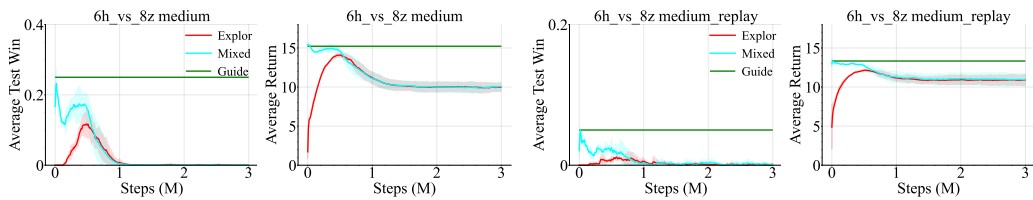

Figure 12: During the training process of JSRL-CFCQL, the performance changes of the guided strategy, exploration strategy, and mixed strategy.

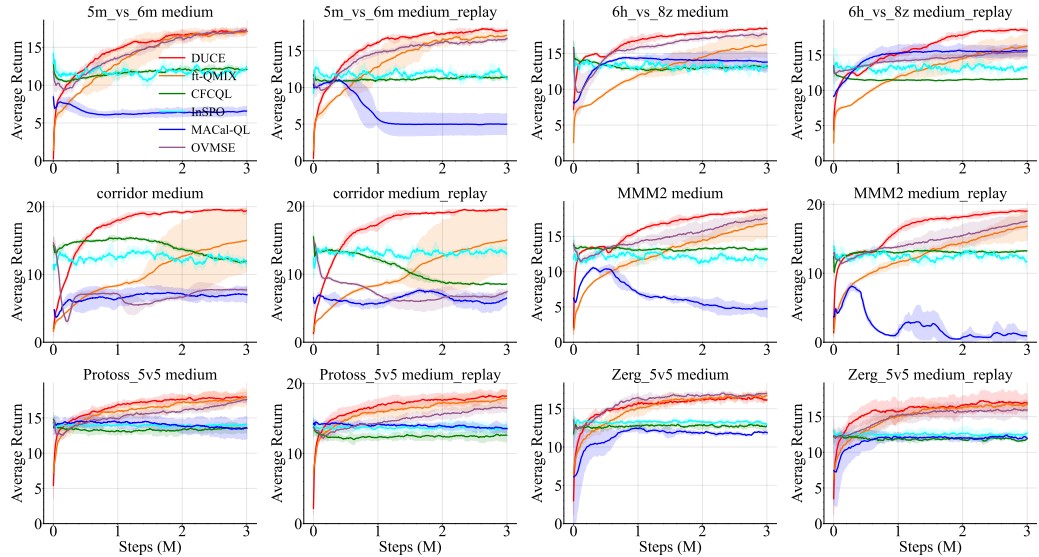

Figure 13: Comparison average test returns with various baselines.

### A.5.2 BASELINE PERFORMANCE WITHOUT USING OFFLINE DATA

In this section, we conduct an additional experiment in which offline data is entirely removed during online training. Specifically, OVMSE, InSPO, CFCQL, and MACal-QL are trained without using any offline data once online learning begins. Fig. 14a reports the performance of these warm-start baselines without offline data and Fig. 14b reports their performance with offline data—on 6h_vs_8z (medium and medium-replay). As shown in Fig. 14a, DUCE significantly outperforms OVMSE, InSPO, CFCQL, MACal-QL, and ft-QMIX. Comparing Fig. 14a and 14b, we observe that removing offline data during fine-tuning causes a substantial performance drop for CFCQL and MACal-QL, leaves InSPO largely unchanged, and leads to a notable degradation for OVMSE on 6h_vs_8z-medium. These results strongly support the conclusion that training solely on online experience during fine-tuning, without retaining offline data, can severely impair the model's ability to preserve what it previously learned from the offline data, thereby causing a performance degradation.

### A.5.3 COMPARISON EXPERIMENTS ON MAMUJOCO

We conduct experiments evaluating DUCE on the widely recognized Multi-agent MuJoCo (MA-MuJoCo) benchmark to demonstrate its applicability to continuous-action multi-agent settings.

**Baselines.** Following our findings on SMAC v1/v2, where ft-QMIX already served as a challenging baseline, we move to continuous-action MARL settings. Since ft-QMIX cannot be directly applied to continuous action spaces, we select MATD3, a widely used actor–critic method for continuous cooperative MARL, as a strong baseline, together with OVMSE. Because OVMSE has no official implementation, we re-implemented OVM and SE based on the MATD3 code in the HARL repository [11]; all other implementation details follow Section 6.1. For fairness, DUCE uses the same

---

[11]https://github.com/PKU-MARL/HARL

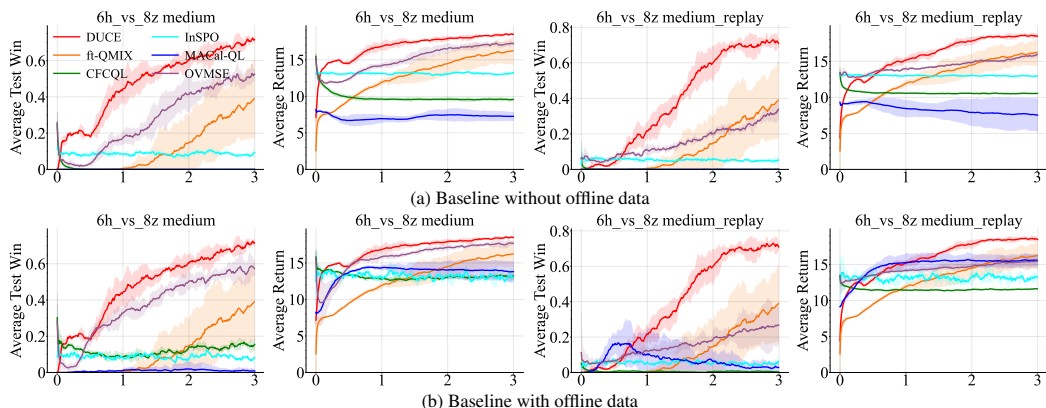

Figure 14: Baseline performance without using offline data.

prior policy as that produced by OVMSE during its offline stage. To adapt DUCE to continuous action spaces, we adopt MATD3 as the underlying MARL algorithm, using its implementation from HARL.

**Experimental setup.** We additionally conducted experiments on the widely recognized MA-MuJoCo benchmark, including challenging scenarios: HalfCheetah-v2 2×3, HalfCheetah-v2 3×2, and HalfCheetah-v2 6×1. Each scenario has two dataset tasks-medium and medium-replay. Following the procedure described in the CFCQL paper, datasets were constructed using the MATD3 algorithm: the medium datasets contain 1000 trajectories, while the medium-replay datasets contain no more than 1000 trajectories (full details in Table 3).

As shown in Fig. 15, DUCE markedly outperforms MATD3 and OVMSE, converging faster and achieving higher asymptotic performance, even though OVMSE benefits from warm starts that initially give it an advantage.

Table 3: Overview of the offline datasets used in our experiments on the MaMujoco HalfCheetah benchmark, summarizing dataset scale, agent composition, space dimensionality, and average returns.

| Instances | | Trajectories | Agents | State dim | Obs dim | Action dim | Average returns |
|---|---|---|---|---|---|---|---|
| HalfCheetah-v2-2x3 | medium | 1000 | 2 | 17 | 19 | 6 | 2.83 |
| | medium_replay | 1000 | 2 | 17 | 19 | 6 | 1.44 |
| HalfCheetah-v2-3x2 | medium | 1000 | 3 | 17 | 20 | 6 | 2.67 |
| | medium_replay | 910 | 3 | 17 | 20 | 6 | 1.28 |
| HalfCheetah-v2-6x1 | medium | 1000 | 6 | 17 | 23 | 6 | 2.87 |
| | medium_replay | 1000 | 6 | 17 | 23 | 6 | 1.85 |

### A.5.4 PERFORMANCE UNDER EXPERT-LEVEL PRIOR GUIDANCE

In this section, we conduct experiments using expert-level priors. Specifically, DUCE-expert denotes DUCE equipped with an expert policy as the prior. As shown in Fig. 16, on 5m_vs_6m and 6h_vs_8z (medium and medium-replay), DUCE-expert shows higher training efficiency than DUCE. This indicates that DUCE is not strongly dependent on the quality of the prior and can also incorporate stronger (expert-level) priors.

### A.5.5 COMPARISON EXPERIMENTS WITH EXTENDED TRAINING STEPS

In this section, we conduct experiments where we extend the training budget of DUCE, ft-QMIX, and OVMSE to 10M environment steps on MMM2, 6h_vs_8z, and Protoss_5v5 (medium and medium-replay). Combining the results from Section 6.2 and this section, DUCE achieves both faster convergence and higher asymptotic performance compared with existing baselines.

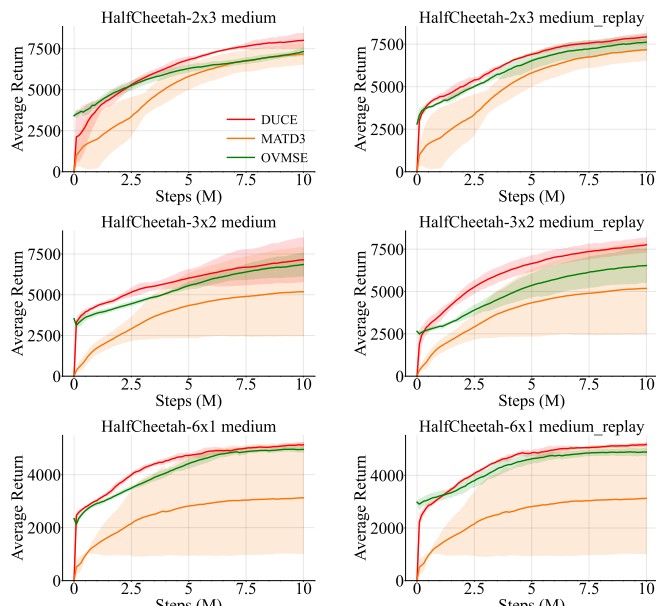

Figure 15: Performance comparison between DUCE and baseline on MaMujoco.

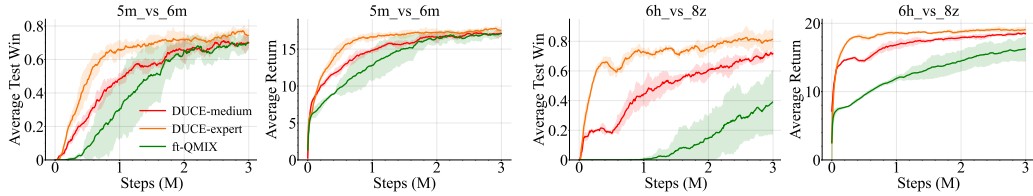

Figure 16: The performance curve of the experiment guided by the strategy trained using DUCE as an expert strategy.

### A.5.6 COMPARISON BETWEEN CURRICULUM-BASED GUIDANCE AND RANDOM-SWITCHING GUIDANCE

In this section, We conduct ablation experiments on prior-guidance mechanisms. In this experiment, we design DUCE-random, which mimics the guidance scheme of JSRL-Random Switching by randomly sampling the number of guide steps in each episode while keeping $p_{\text{guide}} = 1$. As shown in Fig. 18, on 6h_vs_8z and 5m_vs_6m (both medium and medium-replay) show that DUCE-random performs substantially worse than DUCE. The main reason is that although Random Switching increases the diversity of collected samples by randomly varying the guidance length and thereby stabilizes early training, it does not specify a condition under which prior guidance is terminated. As a result, the prior policy continues to dominate even when the MARL agent has surpassed it in the mid-to-late stages of training, which harms overall performance.

Conceptually, these two schemes are distinct. JSRL-Random Switching (and its DUCE-RandomSwitch variant) always follows the guidance policy during the guided segment of an episode; randomness only affects the duration of guidance. In contrast, our $p_{\text{guide}} < 1$ scheme randomizes whether an entire episode is guided or unguided, yielding a mixture of fully guided and unguided episodes. In the presence of an offline regularizer, this distinction is crucial: unguided episodes provide the online policy with room to explore beyond the prior, preventing overly conservative behavior in the early stages of training.

### A.5.7 ADDITIONAL EXPERIMENTS WITH $p_{\text{GUIDE}} = 1$

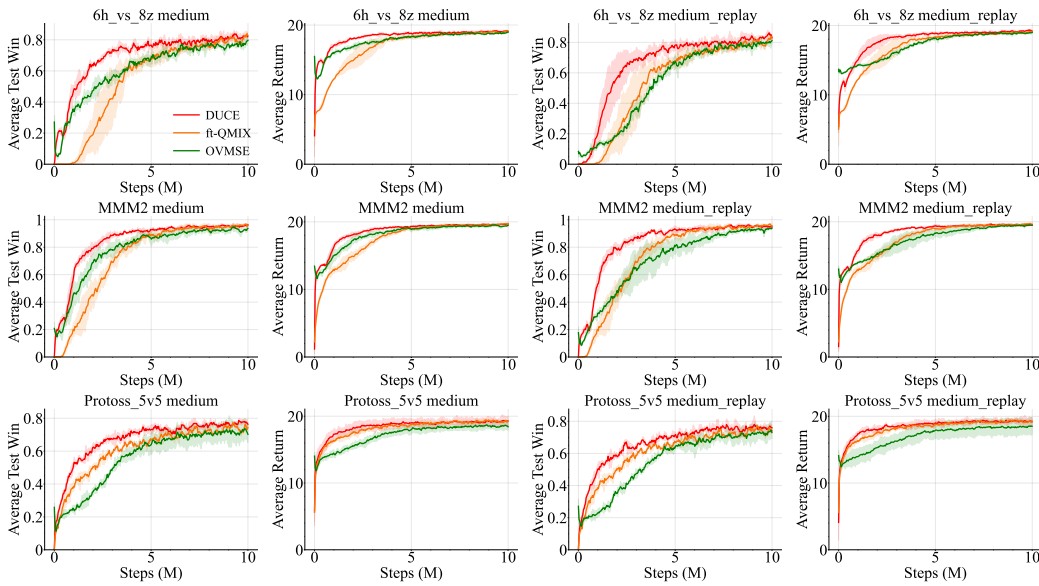

Figure 17: Comparison of training performance between the DUCE method and the baseline method at 10M steps.

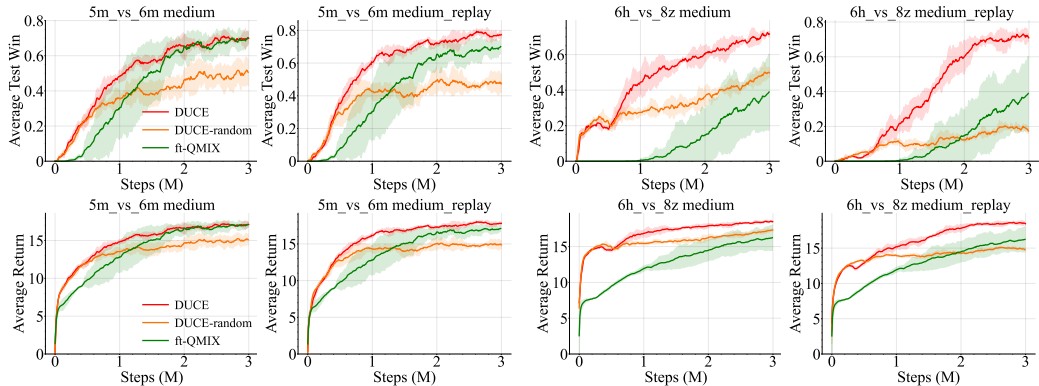

Figure 18: Performance comparison between DUCE random guidance and curriculum guidance.

To illustrate the effect of introducing $p_{\text{guide}}$, we conduct additional experiments on Corridor and 5m_vs_6m under both medium and medium-replay settings. Specifically, we designed a variant DUCE-no $p_{\text{guide}}$, which is identical to DUCE except that we fix $p_{\text{guide}} = 1$. As shown in Fig. 19, DUCE outperforms DUCE-no $p_{\text{guide}}$, especially in the early stage of training, where DUCE achieves faster performance improvement. This indicates that setting can $p_{\text{guide}} = 1$ actually hamper MARL training in these tasks. The main reason is that $p_{\text{guide}} = 1$ causes the early replay buffer to contain an excessively high proportion of prior-guided samples. While this initially benefits conservative updates, the offline regularizer penalizes out-of-distribution Q-values so strongly that exploration becomes restricted, which in turn slows down performance improvement. By contrast, introducing $p_{\text{guide}}$ means that, at the beginning of each episode, we decide whether agents follow pure online exploration or a prior-guided curriculum. This simple mechanism increases the diversity of early collected samples and mitigates the overly restrictive penalties imposed by the conservative regularizer in early training, thereby leading to faster and more stable learning.

### A.5.8 EXPERIMENTS ON DIFFERENT DECAY SCHEDULES FOR THE OFFLINE LOSS WEIGHT

In this section, We conduct a systematic evaluation of different decay schedules in the optimization-based curriculum. Our baseline DUCE uses a linear decay of the regularizer weight from $\alpha_{\text{start}}$

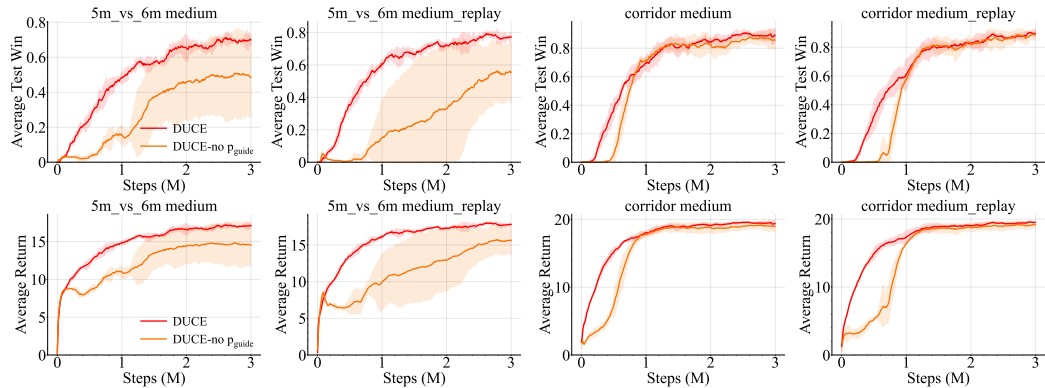

Figure 19: Performance comparison of DUCE without guidance probability.

to 0 over $T_\alpha$ steps. On top of this, we additionally consider: **DUCE-cosine**, which uses a cosine decay schedule $\alpha_t = \frac{1}{2}(\alpha_{\text{start}})(1 + \cos(\pi t/T_\alpha))$. This is a monotone, non-periodic half-cosine that smoothly interpolates from $\alpha_{\text{start}}$ to 0: the weight decreases once and does not rise again, so it does not introduce oscillatory behavior. It also introduces no additional hyperparameters beyond $\alpha_{\text{start}}$, and $T_\alpha$, i.e., the same parameters used by the linear schedule. Cosine decay is a standard choice in deep reinforcement learning for annealing learning rates or exploration parameters, and we adopt it here as a representative smooth alternative to linear decay. **DUCE-fixed**, which keeps the regularizer weight fixed at $\alpha_{\text{start}}$ throughout training (no decay).

In all experiments, we use the same $(\alpha_{\text{start}}, T_\alpha)$ across the different schedules, without additional tuning for each schedule. As shown in Fig. 20 results on 6h_vs_8z and 5m_vs_6m (medium and medium-replay) show that DUCE and DUCE-cosine behave very similarly: both achieve fast and stable learning and reach comparable final performance. In contrast, DUCE-fixed fails to learn effective policies, consistent with our Observation in Sec. 4: while adding an offline loss mitigates distribution shift and improves early-stage stability, maintaining a large penalty throughout training restricts exploration, confines collected samples to a narrow region of the state space, and eventually leads to overfitting.

These results suggest that the key property is simply that the regularizer weight decays over time; the exact functional form of the decay (linear vs. cosine) is not critical as long as it is monotone. For this reason, and for simplicity and ease of reproduction, we retain the linear decay schedule as the default choice in DUCE.

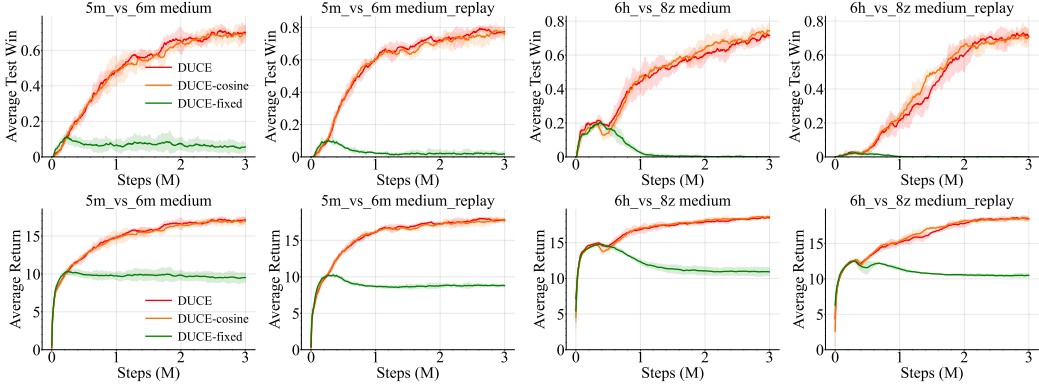

Figure 20: Performance comparison of different offline loss weight decay methods.

### A.5.9 EXPERIMENTS ON PER-AGENT CURRICULUM DESIGNS

For per-agent curriculum ablations, we designed DUCE-G1 and DUCE-G2 as minimal yet representative instantiations of the two ideas: (i) different agents may be suitable for different strategies

for determining guide-step sequences; (ii) different agents may be suitable for different curriculum parameters. In DUCE-G1, the first half of the agents follows a curriculum-based guidance horizon, whereas the second half samples the guide steps randomly at the episode level. In DUCE-G2, the first half follows a curriculum with 10 stages, while the second half follows one with 20 stages.

Why DUCE-G1 mixes curriculum-based and random guide-step strategies. In JSRL, there are essentially two canonical mechanisms for controlling the guide-step sequence: (a) a curriculum-based schedule that gradually shortens the guidance horizon; and (b) a purely random switching strategy. DUCE-G1 therefore partitions the agents into two sub-groups, with one group following the curriculum-based guidance horizon and the other group sampling guide steps randomly at the episode level. This construction is intentionally simple: it directly realizes point (i) without introducing additional, ad-hoc strategies that do not appear in the original JSRL framework. In other words, DUCE-G1 is a clean and interpretable test case of "different agents follow different guide-step strategies", built solely from the two standard mechanisms already present in JSRL. More sophisticated variants (e.g., many different strategies per agent or complex learned scheduling rules) would add a large number of design choices and hyper-parameters, making it harder to attribute any performance change to the core idea of per-agent strategy heterogeneity.

Why DUCE-G2 varies the number of curriculum stages as the key curriculum parameter. For point (ii), we need a parameter that genuinely characterizes the curriculum, rather than a peripheral hyper-parameter. In our curriculum, the number of stages controls how finely the guidance horizon is annealed: more stages mean a finer-grained, slower decay of prior guidance, while fewer stages mean a coarser, more aggressive decay. We therefore let different sub-groups of agents share the same curriculum structure but use different numbers of stages (e.g., 10 vs. 20). This directly instantiates "different agents have different curriculum parameters" in a way that is both semantically meaningful (it changes the granularity of prior guidance) and easy to interpret. Again, we deliberately avoid an overly elaborate design (e.g., tuning many different parameters per agent), because our goal is to test the principle of agent-wise curriculum heterogeneity, not to engineer a highly optimized heterogeneous scheme.

As shown in Fig. 21, on 6h_vs_8z-medium and 6h_vs_8z-medium-replay, both DUCE-G1 and DUCE-G2 exhibit slower early learning and slightly lower final win rates than the original DUCE with a single global curriculum, and they do not bring systematic improvements to ft-QMIX. In other words, making the guidance schedule heterogeneous across agents tends to increase training variance without yielding consistent gains in our CTDE value-factorization setting.

Based on these results, we chose a single global guidance schedule, which is already validated by our ablations as effective for improving MARL training efficiency.

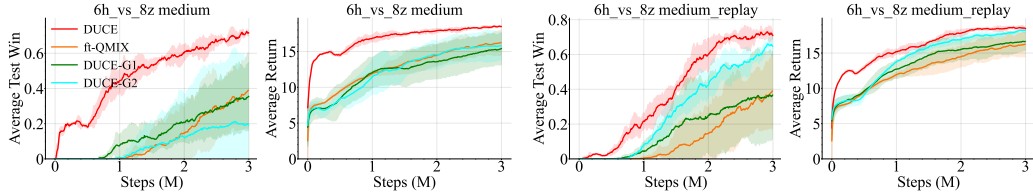

Figure 21: Performance comparison of strategy guidance conducted by different intelligent agent groups.

### A.5.10 VISUALIZATION OF STATE DISTRIBUTIONS VISITED BY THE PRIOR AND ONLINE POLICIES

To more clearly analyze the improvements introduced by DUCE, we visualize the state distributions visited by the prior and online policies. Specifically, we apply t-SNE to embed the global states into a three-dimensional space and plot the states visited by the prior and by the online policy at the beginning and at the end of training.

Fig. 22 is organized into two blocks: the *left* block corresponds to the beginning of training, and the *right* block corresponds to the end of training. In each block, the top-left subfigure shows the overall 3D t-SNE embedding of the visited states (the main object of interest), while the remaining three

subfigures depict different projection distributions, respectively, to help interpret the structure of the state space.

From the left block (before training), the state distributions visited by the prior and online policies differ. Comparing the left and right blocks, we observe that the online policy's distribution gradually moves across the embedding space toward and through the region occupied by the prior, and eventually expands into a broader area beyond the prior's coverage. This evolution indicates that DUCE's dual-track curriculum uses the prior policy as a directional guide for exploration—steering the online policy toward informative regions of the state space while allowing it to further explore beyond the prior's support, thereby improving training efficiency.

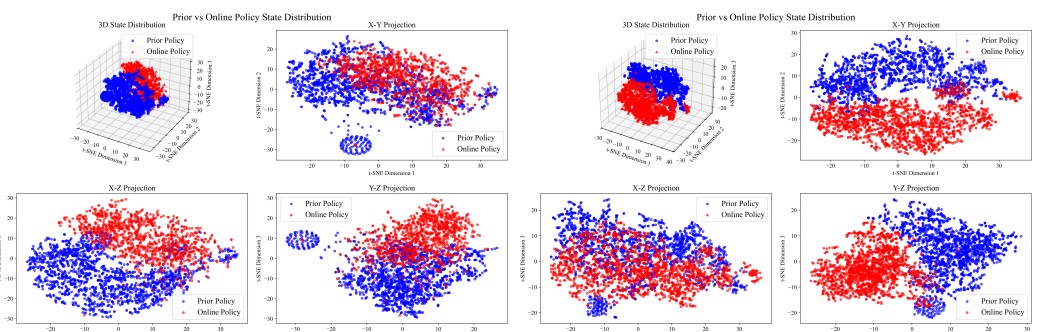

Figure 22: The distribution difference between the prior strategy and the online strategy access states is depicted using t-SNE. The left side shows the distribution before training, while the right side shows the distribution after training.

A.5.11 THE PERFORMANCE OF A PRIORI STRATEGIES

**1) Models trained offline using CFCQL**   The prior policies used in our experiments were uniformly trained offline with CFCQL for a total of 3 million environment steps across different tasks. Table 4 summarizes the average test win-rates and average episodic returns with standard deviation as error bars.

Table 4: Average win-rate and reward of prior policies trained offline using offline datasets (with standard deviation as error bar)

| Instances | | Average Test Win | Average Return |
|---|---|---|---|
| 5m_vs_6m | medium | $0.20 \pm 0.19$ | $11.66 \pm 1.91$ |
| | medium_replay | $0.23 \pm 0.21$ | $11.90 \pm 2.23$ |
| 6h_vs_8z | medium | $0.25 \pm 0.25$ | $15.22 \pm 2.09$ |
| | medium_replay | $0.05 \pm 0.15$ | $13.33 \pm 1.42$ |
| corridor | medium | $0.35 \pm 0.20$ | $14.46 \pm 1.71$ |
| | medium_replay | $0.20 \pm 0.19$ | $14.16 \pm 1.60$ |
| MMM2 | medium | $0.17 \pm 0.27$ | $13.50 \pm 2.06$ |
| | medium_replay | $0.15 \pm 0.17$ | $12.17 \pm 1.84$ |
| Protoss_5v5 | medium | $0.21 \pm 0.08$ | $14.06 \pm 0.83$ |
| | medium_replay | $0.23 \pm 0.08$ | $14.14 \pm 0.81$ |
| Zerg_5v5 | medium | $0.32 \pm 0.09$ | $12.81 \pm 1.15$ |
| | medium_replay | $0.33 \pm 0.08$ | $12.91 \pm 0.98$ |

**2) Models trained with ft-QMIX combined with a linear network**   The linear model employed for guidance in this paper is trained using ft-QMIX, which differs from the ft-QMIX utilized in the baseline. Specifically, we replaced the original RNN with a 3-layer fully connected layer while maintaining the same dimensionality. The training process is terminated and the model is saved as

the guidance model when its win-rate becomes comparable to that of the offline-trained prior policy. The win-rate of the model for the 6h_vs_8z scenario is 0.2, and that for the Zerg_5v5 is 0.35.

**3) Rules formulated using LLMs**   Given that SMAC constitutes a well-established and classical environment—with the two scenarios, namely 6h_vs_8z and Zerg_5v5, being extensively utilized—specific descriptions of the environment are deemed unnecessary when formulating prompt engineering for LLMs. Instead, it is sufficient to clearly define the input and output parameters, while providing the corresponding state information to the LLM.

In our experiments, we considered adopting encoded global states as the input and decision-making actions as the output. Concurrently, within the prompts, we specified the indices of key states, such as the health points (HP) of allied units, relative distances between allied units, the HP of enemy units, and relative distances between enemy units. The detailed design of the prompts is illustrated in Fig. 23.

Naturally, the outputs of large language models (LLMs) are not entirely accurate. Furthermore, our prompts were relatively concise; through simple debugging to rectify minor bugs, we obtained the rules utilized in the experiment. Owing to the strong robustness of our algorithm, we hypothesize that even guidance via the simplest set of rules would be effective. Consistent with our expectations, a set of the most basic rules, randomly generated by the LLM, was also able to accelerate the convergence rate to a certain extent. The core logic of these rules revolves around "focused fire": prioritizing attacks on enemy units with the lowest HP. If the unit's own HP is the lowest among allied forces, it will retreat. Due to the absence of micro-operations such as kiting and formation maneuvering, when applied to the scenarios of 6h_vs_8z and Zerg_5v5, the average reward achieved was approximately 7 (slightly higher than that of the random strategy), with nearly no winning rate. Detailed strategies are presented in Algorithm 1.

---

**Algorithm 1:** ConcentrateFireGlobal: Global-State-Based Focus Fire Policy

---

**Input:** global $state$, $avail\_actions$
**Output:** $action$
Extract $ally\_health$, $enemy\_health$;
$enemy\_alive \leftarrow (enemy\_health > 0)$;
Select lowest-health enemy $target\_enemy$; if none alive, set $\emptyset$;
**for** *each agent $a$* **do**
  // Agents operate in parallel
  **if** $a$ *is dead* **then**
   | $action[a] \leftarrow 0$ ;                                    // dead → action 0
  **end**
  **else**
    **if** $target\_enemy \neq \emptyset$ **then**
     | $action[a] \leftarrow$ attack $target\_enemy$;
    **end**
    **else**
      **if** $a$ *is lowest-health ally* **then**
       | $action[a] \leftarrow$ move away from enemy;
      **end**
      **else**
       | $action[a] \leftarrow$ move toward enemy;
      **end**
    **end**
    **if** $action[a] \notin avail\_actions[a]$ **then**
     | $action[a] \leftarrow 1$ ;                                   // stop
    **end**
  **end**
**end**
**return** $action$

---

## A.6 THEORETICAL ANALYSIS

**Prompt:**
**# You are a strategic analyst proficient in StarCraft Multi-Agent Challenge (SMAC), tasked with formulating a set of battle rules for players in the 6h_vs_8z or Zerg_5v5 match-up scenarios.**
**# Requirement**
You need to implement the ConcentrateFireGlobal class and the get_guide_actions method code, generating concise function input and output rules for SMAC (StarCraft Multi-Agent Challenge), covering core definitions, formats, and associated logic, highlighting SMAC adaptability. Specific requirements are as follows:
**1.** Function positioning
The get_guide_actions function of ConcentrateFireGlobal provides focus-fire guidance actions for SMAC multi-agent systems, adapting to combat modes with arbitrary n_agents (the number of our agents) and n_enemies (the number of enemy agents). It links the SMAC action space with the global state mechanism.
**2.** Input parameter rules (divided into initialization + method parameters)
**(1)** Class initialization parameters (init)
args: Stores SMAC configuration, dictionary / custom object type.
n_agents: The number of our agents.
n_enemies: Number of enemy agents.
ally_feats_dim: The state dimension of a single friendly agent.
enemy_feats_dim: The state dimension of a single enemy agent.
**(2)** Parameters of the get_guide_actions method
agent_inputs: Local observations of agents (tensor, e.g., (batch_size, n_agents, obs_dim)).
t: The current episode number (integer, $0 \le t <$ episode_limit, used to extract the current episode data).
ep_batch: Core match data batch (dictionary):
ep_batch["state"]: Global state (tensor, (batch_size, episode_limit, total_dimension)).
ep_batch["avail_actions"]: Available actions (tensor, (batch_size, episode_limit, n_agents, action_dimension)), where 1 = available, 0 = unavailable.
**3.** Output result rules
Output `chosen_actions` (a PyTorch long tensor of shape `(batch_size, n_agents)`), where each element represents the action encoding of an agent, with the following meanings:
Action 0: Output only when the agent dies (ally_health $\le 0$), corresponding to no operation in SMAC.
Action 1: The agent survives but the action is invalid, corresponding to SMAC "Stop".
Action 2-5: Movement action (2 = down, 3 = up, 4 = right, 5 = left).
Action 6+: Attack action (6 + enemy ID), generated only when there are enemies alive.
Note that all actions should be checked by avail_actions at the end.
**# Status description**
Global state, where friendly features:
ally_state[al_id, 0] = (al_unit.health / al_unit.health_max)
ally_state[al_id, 2] = (x - center_x) / self.max_distance_x
ally_state[al_id, 3] = (y - center_y) / self.max_distance_y
the enemy's characteristics are as follows:
enemy_state[e_id, 0] = (e_unit.health / e_unit.health_max)
enemy_state[e_id, 1] = (x - center_x) / self.max_distance_x
enemy_state[e_id, 2] = (y - center_y) / self.max_distance_y

Figure 23: Prompt design for generating rules of LLMs, showing key input state indices and the corresponding action outputs used for decision-making in SMAC scenarios.

We provide a theoretical justification for DUCE. Our goal is to explain why, in cooperative MARL, (i) naive non-optimistic exploration such as $\epsilon$-greedy ft-QMIX can require exponential samples in the horizon, while (ii) the dual-track curriculum in DUCE reduces the sample complexity to a polynomial dependence, under suitable assumptions on the prior policy.

**Centralized view of Dec-POMDPs.** We adopt the standard CTDE view and regard the joint action-value function $Q_{\text{tot}}(s, a)$ of ft-QMIX as the $Q$-function of a single "centralized" agent acting in an MDP $M^c = (S, \mathcal{A}, P, R, p_0, \gamma, H)$, where $\mathcal{A}$ is the joint action space. The decentralized policies only affect how the centralized joint policy factorizes at execution time, and do not affect the following analysis. In this view, DUCE and other baselines differ only in how they collect data and optimize $Q_{\text{tot}}$.

**Suboptimality and sample complexity.** For any joint policy $\pi$ in $M^c$, let

$$J(\pi) \triangleq \mathbb{E}_{s_0 \sim p_0}[V^\pi(s_0)], \quad \Delta(\pi) \triangleq \mathbb{E}_{s_0 \sim p_0}\big[V^\star(s_0) - V^\pi(s_0)\big]$$

denote the expected return and suboptimality, respectively. We say that an algorithm attains sample complexity $T(\varepsilon)$ if, after collecting at most $T(\varepsilon)$ environment steps, it outputs a policy $\hat{\pi}$ with $\Delta(\hat{\pi}) \leq \varepsilon$.

**Exponential hardness of naive MARL exploration.** Consider applying ft-QMIX with $0$-initialized $Q_{\text{tot}}$ and an $\epsilon$-greedy exploration strategy from scratch (no prior, no curriculum). By viewing $M^c$ as a standard episodic MDP, the combination-lock construction used in the JSRL analysis implies that there exists a cooperative Dec-POMDP (hence a centralized MDP $M^c$) for which any such non-optimistic method must incur a sample complexity that is exponential in the horizon $H$ in order to achieve $\Delta(\hat{\pi}) < 1/2$. Intuitively, without guidance, the joint agent needs to guess the correct sequence of joint actions, whose success probability can scale as $2^{-H}$. This shows that, even in cooperative MARL, naive $\epsilon$-greedy exploration can be exponentially hard in $H$, motivating the need for prior-guided curricula.

### A.6.1 ASSUMPTIONS ON THE PRIOR JOINT POLICY

Let $d_h^\pi$ be the state visitation distribution at time $h$ under a joint policy $\pi$ in $M^c$. Following the JSRL analysis, we assume that the prior joint policy $\pi_g$ covers all "good" states visited by the optimal policy under a feature representation.

**Assumption A.1** (Quality of prior joint policy). Assume that there exists a feature map $\phi : S \to \mathbb{R}^d$ such that, for any policy $\pi$, both $Q_{\text{tot}}^\pi(s, a)$ and $\pi(a \mid s)$ depend on $s$ only through $\phi(s)$. Moreover, there exists a constant $C \geq 1$ such that

$$\sup_{s,h} \frac{d_h^{\pi^\star}(\phi(s))}{d_h^{\pi_g}(\phi(s))} \leq C.$$

That is, in the feature space, the prior joint policy $\pi_g$ visits every state that the optimal policy visits, up to a bounded density ratio $C$.

This assumption is strictly weaker than requiring that the prior visits every optimal state–action pair; it only requires coverage of optimal states in feature space.

### A.6.2 TASK-DIFFICULTY CURRICULUM: FROM EXPONENTIAL TO POLYNOMIAL

To isolate the role of DUCE's externally configured task-difficulty curriculum, we first consider a simplified variant, denoted DUCE-TASK, which sets $p_{\text{guide}} = 1$ and $\alpha(t) \equiv 0$ (no offline regularizer). In this case, DUCE exactly reduces to JSRL applied to the centralized MDP $M^c$, where the guide policy is $\pi_g$, the exploration policy is the joint policy of ft-QMIX, and the sequence of guide steps $(H_1, \ldots, H_n)$ is given by the curriculum stages with decreasing $h$.

Under this reduction, DUCE-TASK coincides with the Policy Search by Dynamic Programming(PSDP)-style variant analyzed in prior work (Bagnell et al., 2003; Uchendu et al., 2023), and we obtain a polynomial sample complexity bound.

**Theorem 1** (Polynomial sample complexity of DUCE-TASK). *Consider tabular MARL under the centralized MDP view with $|S|$ joint states and $|\mathcal{A}|$ joint actions. Suppose Assumption A.1 holds, ft-QMIX uses an $\epsilon$-greedy exploration policy, and the task-difficulty curriculum in DUCE chooses the guide steps $(H_1, \ldots, H_n)$ in a backward schedule that only moves to the next stage when the performance of the mixed policy crosses a threshold $\beta \in (0,1)$. Then there exists a constant $c > 0$ such that, with probability at least $1 - \delta$, after collecting $T$ samples,*

$$\Delta(\hat{\pi}) \leq c\, C\, H^{5/2}|S|^{1/2}|\mathcal{A}|^{1/2}T^{-1/2},$$

*where $\hat{\pi}$ is the final exploration policy of* DUCE-TASK. *In particular, for any $\varepsilon > 0$, the required sample complexity $T(\varepsilon)$ is polynomial in $(H, |S|, |\mathcal{A}|, C, 1/\varepsilon)$.*

**Proof sketch.** Under the centralized view, the mixed policy in DUCE rolls in with $\pi_g$ for $h$ steps, then rolls out the exploration policy for the remaining $H - h$ steps. At each curriculum stage, DUCE trains the exploration policy only on the suffix starting from step $h + 1$, while the prefix is controlled by $\pi_g$. This is exactly the PSDP-style update analyzed in prior work: the exploration policy only needs to optimize its behavior on the suffix, while benefiting from the state distribution induced by $\pi_g$ on the prefix.

Assumption A.1 ensures that this distribution provides sufficient coverage of all states visited by the optimal policy in the feature space. Therefore, the upper bound on suboptimality for PSDP/JSRL directly applies to DUCE-TASK, up to replacing the number of actions by the joint action space size $|\mathcal{A}|$. The resulting bound has a polynomial dependence on $H$ and $1/\varepsilon$, contrasting with the exponential lower bound for naive non-optimistic exploration.

### A.6.3 EFFECT OF THE OPTIMIZATION-BASED CURRICULUM

We incorporate DUCE's internally evolved optimization-based curriculum, which augments the ft-QMIX online loss $L_{\text{QMIX}}$ with a decaying offline regularizer $\alpha(t)L_{\text{CFCQL}}$:

$$L_t = L_{\text{QMIX},t} + \alpha(t)L_{\text{CFCQL},t},$$

where $\alpha(t)$ decreases linearly from $\alpha_{\text{start}}$ to $0$ over $T_\alpha$ gradient steps. This yields the full DUCE algorithm.

We model the update of $Q_{\text{tot}}$ as a stochastic approximation process with an additional vanishing perturbation induced by the offline loss.

**Assumption A.2** (Bounded offline regularizer and decay). Assume that the offline loss is uniformly bounded in expectation, i.e. there exists $B > 0$ such that $\mathbb{E}[L_{\text{CFCQL},t}] \leq B$ for all $t$, and that the weight schedule $\alpha(t)$ is non-increasing, $\lim_{t \to \infty} \alpha(t) = 0$, and satisfies $\sum_{t=1}^{\infty} \alpha(t)\eta_t < \infty$, where $\{\eta_t\}$ are the learning rates used in the stochastic gradient updates of $Q_{\text{tot}}$.

Under Assumption A.2, the offline term behaves as a bounded vanishing perturbation in the stochastic approximation dynamics.

**Theorem 2** (DUCE preserves polynomial sample complexity). *Under the conditions of Theorem 1 and Assumption A.2, the full DUCE algorithm with dual-track curricula enjoys the same polynomial sample complexity order as* DUCE-TASK. *In particular, there exist constants $c_1, c_2 > 0$ such that, with probability at least $1 - \delta$, the final policy $\hat{\pi}$ output by DUCE satisfies*

$$\Delta(\hat{\pi}) \leq c_1\, C\, H^{5/2}|S|^{1/2}|\mathcal{A}|^{1/2}T^{-1/2} + c_2 \max_{t \leq T} \alpha(t),$$

*so that the second term vanishes as training proceeds and the asymptotic sample complexity remains polynomial in $(H, |S|, |\mathcal{A}|, C, 1/\varepsilon)$.*

**Proof sketch.** The first term in the bound follows from the analysis of DUCE-TASK. The additional offline regularizer $\alpha(t)L_{\text{CFCQL},t}$ only affects the target values used in the Bellman updates for $Q_{\text{tot}}$. Under Assumption A.2, this perturbation is uniformly bounded and its effective contribution to the stochastic approximation dynamics decays to zero.

Classical results on stochastic approximation with vanishing perturbations then imply that the iterates of $Q_{\text{tot}}$ under DUCE converge to the same set of limit points as those of DUCE-TASK, up to

a bias proportional to $\max_{t \leq T} \alpha(t)$. Consequently, the asymptotic suboptimality and sample complexity orders are unchanged. At the same time, the offline regularizer reduces estimation error on prior-generated data and mitigates distribution shift in early stages, which matches our empirical observation that DUCE improves both sample efficiency and performance compared to ft-QMIX and prior-guided baselines.

