# OpenReview forum: "One Policy Learns Them All: Synergizing  Prior-Guided Exploitation and Online Exploration in Curriculum Based MARL"
_ICLR.cc/2026/Conference — Submitted to ICLR 2026_

### Official Review · Reviewer_wwsc · 2025-10-21

**Soundness:** 1
**Presentation:** 2
**Contribution:** 2
**Rating:** 2
**Confidence:** 4

**Summary:**

The main idea of the paper is combining some approaches from Offline Multi-Agent Reinforcement Learning (primarily CFCQL) and guided policy learning (primarily JSRL) to develop a more sample efficient MARL framework in an online setting, where the policy is guided using both a regularization loss based on an expert policy and jump-starting from intermediate states obtained by the same expert policy to improve exploration. Both these guidances are scheduled to decrease over the course of training.

I would summarize the main contributions compared to prior work as:
1. Applying the previously suggested JSRL in a multi-agent setting.
2. Specifically, the combination of JSRL and the previously suggested CFCQL.
3. Using a linear schedule for previously constant weight of the regularization term in the existing CFCQL loss for offline-MARL.
4. Introducing a probability of guidance p_guide in the JSRL algorithm, which at the start of an episode determines whether guidance shall be used at all. The default for "vanilla" JSRL would be p_guide=1.

**Strengths:**

The paper presents a novel framework for prior-guided multi-agent reinforcement learning which combines an expert policy both to regularize the learned policy towards a desired behavior and help learning by jump starting trajectories. The combination of these methods is well motivated and the proposed scheduling reasonable. The overall motivation is sound and the research topic of interest to the RL community.

**Weaknesses:**

## Regarding main contributions

The abstract and sections 1, 6, A4 claim higher asymptotic performance of the proposed method compared to SOTA methods. However, figures 4 and 10 show non-convergence of the performance of other methods such as ft-QMIX.

The application of JSRL in a multi-agent setting, even in the specific context of SMAC, is not novel, see "SC-MAIRL: Semi-Centralized Multi-Agent Imitation Reinforcement Learning", Brackett et al. This literature is missing.

Section 6.3 claims for p_guide that "When the values exceed 0.7, substantial performance fluctuations emerge during the early training phase." However, in Figure 6, the blue line for p_guide=1 is at all stages of training the best or close-to-best performing variant. Since p_guide=1 is from my understanding the base setting for JSRL, I do not see where it is shown that this contribution is beneficial. Furthermore, the JSRL paper also contains an ablation called "JSRL-Random Switching", where the guidance length is randomly varied and not monotonically decreasing, which conceptually sound similar to the p_guide introduced here.

From my understanding, the primary difference to the standard CFCQL loss is linearly scheduling the weight alpha of the regularization, which is constant in CFCQL. I consider both the design choice as a constant in an offline setting for the original CFCQL algorithm and decaying it in an online setting as done in the proposed paper well motivated. Non-constant weighting of a regularizing loss based on for example an expert policy in an offline-to-online setting is however not a novel contribution, see as one example "Adaptive Behavior Cloning Regularization for Stable Offline-To-Online RL", Zhao et al.
Given that introducing a schedule to a regularization loss is of limited novelty, and is only novel in the context of CFCQL, I would have expected a strong and convincing evaluation of a variety of schedules. However, the scheduling is only evaluated in one scenario in figure 8, only the parameters alpha_start and T_alpha are varied and no comparison is made to other schedules, despite hinting at different schedules in the appendix. Furthermore, the authors themselves write that the length of the decay period T_alpha has limited impact on the overall training efficiency and that the choice of alpha_start has limited influence on the overall training efficiency (both section 6.4). This raises the question how important this contribution is.

In general, the evaluation is performed on different scenarios within the SMAC environment. However, I consider this somewhat limited. In particular, the authors only evaluate on discrete action spaces. Previous work such as CFCQL has used SMAC, MPE and MA-MuJoCo in their evaluations.

## Regarding presentation and notation

In section 3, the definition of domain of pi is given as o_t x a_t. However, o_t and a_t are not defined.

Observation II claims that Fig. 2c shows Q_tot of JSRL-CFCQL to "exhibit sharp early fluctuations before stabilizing", but JSRL-QMIX has much larger shaded area whereas JSRL-CFCQL is as narrow as ft-QMIX?

All regarding Equation (1), which is the core loss introduced by the authors:
It is not explained what a_-i notation represents. From CFCQL I expect this to be the joint action without the action of agent i, but this should be clearly expressed.
$E_{a \sim \eta}[Q(s,a)]$, a should likely be bold because I have interpreted it to be a joint action.
The notation $E_{\pi_i} (\pi_i(s)/\eta_i(s))$ is unclear. My best guess is $E_{a \sim \pi_i(\cdot|s)} (\pi_i(a|s) / \eta_i(a|s))$. If this is correct, there are several issues which I do not see adressed:
First of all, since samples are drawn from $\pi$, but division happens through probability of $\eta$, if any sample is never selected by $\eta$, this expected value is not defined. Secondly, without further tricks, I would expect that one must be able to evaluate $\eta(a|s)$. Being able to calculate $\eta(a|s)$ is a stronger requirement than being able to sample $a \sim \eta(\cdot|s)$, the general requirement for a (black box) stochastic policy. Yet, the paper claims that it addresses "prior policies without assumptions on their form or structure". Maybe this statement should be relaxed, or the authors should clarify why this is not a concern. I am aware the CFCQL mitigates this by training a VAE to mimic the behavior policy, but whatever is done by the authors should be explicitly stated.

**Questions:**

I would ask the authors to clarify my questions also mentioned in Weaknesses regarding handling behavior policies for which a probability density function is not straightforward to calculate. I would appreciate if they could further clarify the points regarding notation.

*Suggestions regarding current weaknesses*:

I recommend repeating the main experiments with higher step number such that performance of all methods saturates (or deteriorates if this happens) or removing the claim regarding better asymptotic performance.

I suggest citing the work mentioned in Weaknesses regarding JSRL in a MARL context and discussing the differences to this work.

It would be interesting to further analyze how similar the previously suggested JSRL-Random Switching curriculum is to the proposed p_guide<1, since both introduce random ordering and in general extend the analysis of this contribution.

Similarly, I would recommend extending the analysis on different schedules for the regularization loss, especially compared to a variety of constant factors, and clearly showing the benefit on a variety of tasks as this is a claimed contribution.

Given the broad claims of applicability, especially that the proposed framework is defined on arbitrary action spaces and not just discrete ones, I would suggest further analysis to its benefits on continuous action spaces, or since this may be difficult within the time frame, clarify the limitations in evaluation.

---

> ### Author Response · Authors · 2025-11-25
> **Response to Reviewer wwsc (1/7)**
>
> We sincerely thank the reviewer for the very careful reading and many detailed comments. We have revised both the paper and our empirical analysis accordingly. Below we address each point (Weaknesses W1–W8 and Questions Q1–Q6) and highlight the corresponding changes in the manuscript.
>
> ## Weaknesses
>
> ### Regarding main contributions
>
> > **W1:** The abstract and sections 1, 6, A4 claim higher asymptotic performance  of the proposed method compared to SOTA methods. However, figures 4 and  10 show non-convergence of the performance of other methods such as  ft-QMIX.
>
> **Response to W1:** Given the tight time constraints and our limited computational resources, we carefully examined the main experiments in Section 6.2 to identify the maps on which some algorithms had not converged—specifically, ft-QMIX and OVMSE on medium and medium-replay tasks of  MMM2 , 6h_vs_8z, and Protoss 5v5. Accordingly, we have added experiments in Appendix A.5.5 where we extend the training budget of DUCE, ft-QMIX, and OVMSE to 10M environment steps on these tasks. Combining the results from Section 6.2 and Appendix A.5.5, DUCE achieves both faster convergence and higher asymptotic performance compared with existing baselines.

---

> > ### Author Response · Authors · 2025-11-27
> >
> > Dear Reviewer wwsc,
> >
> > As the author–reviewer discussion period will conclude soon, we would be grateful if you could kindly take a moment to look over our responses to your comments. If you have any additional questions or feedback, we would be very happy to address them while the discussion window is still open.
> >
> > Thank you very much for your time and consideration.
> >
> > Best regards,
> > The Authors

---

> ### Author Response · Authors · 2025-11-25
> **Response to Reviewer wwsc (2/7)**
>
> > **W2:** The application of JSRL in a multi-agent setting, even in the specific  context of SMAC, is not novel, see "SC-MAIRL: Semi-Centralized  Multi-Agent Imitation Reinforcement Learning", Brackett et al. This  literature is missing.
>
> **Response to W2:**
>
> We have discussed SC-MAIRL in the related-work section. SC-MAIRL focuses on a **semi-centralized execution** setting, whereas we study the more challenging **decentralized execution** scenario. Moreover, SC-MAIRL does not use JSRL; instead, it employs a comparatively coarse mechanism for teacher-guided MARL, which suffers from rigid guidance schedules and requires high-quality prior policies. Moreover, our method does not simply apply JSRL to the multi-agent setting. The detailed analysis is as follows:
>
> 1. **Guidance mechanism.**
>
>    Compared with JSRL, SC-MAIRL adopts a “teaching window” mechanism to imitate the teacher policy, but this design has two notable limitations:  (i) Algorithm 1 of SC-MAIRL shows that the teacher’s guidance length is fixed and does not account for teacher performance. (ii) SC-MAIRL reduces the teaching window by exactly one step every fixed number of iterations—a rigid schedule that ignores the MARL agent’s learning progress and cannot guarantee stable improvement as the guidance decays.
>
> 2. **Quality of the prior policy.**
>
>    Compared with JSRL, SC-MAIRL relies on a **high-quality pre-trained teacher** to guide an untrained student policy. Specifically, as shown in Table 3 of the SC-MAIRL paper, this teacher is obtained by training SC-MARL from scratch for 14M steps, achieving performance close to an optimal policy. From an algorithmic perspective, SC-MAIRL clearly assumes a high-quality prior: its guidance window is fixed at 150 steps, which implicitly requires the prior policy’s average episode length to exceed 150 steps. This is a substantial requirement for the teacher policy used in its experiments. Both the algorithm design and experimental setup indicate that SC-MAIRL depends heavily on a strong prior, which is unrealistic in real-world settings where high-quality expert policies are extremely difficult to obtain.
>
> 3. **DUCE is not a simple application of JSRL to the multi-agent setting.**
>
>    Preliminary experiments on JSRL-QMIX and JSRL-CFCQL show that **simple combinations of JSRL with standard MARL backbones are insufficient to solve the cold-start MARL training problem**. Concretely, we observe two failure modes:  **(i)** Extending JSRL to cooperative MARL with ft-QMIX as the backbone (JSRL-QMIX) yields worse training efficiency than ft-QMIX itself and suffers from severe overestimation of $Q_{\text{tot}}$ under prior guidance due to distributional shift (Section 4, Observation I); and **(ii)** Augmenting JSRL-QMIX with an offline regularizer based on CFCQL  (JSRL-CFCQL) temporarily improves early performance, but its excessive penalties constrain exploration, confine collected samples to a narrow local region of the state space, inflate in-distribution Q-values, and ultimately lead to overfitting (Section 4, Observation II).
>
>    Motivated by these observations, we propose DUCE, a dual-track curriculum tailored to cold-start MARL with prior-guided exploration. **The core novelty of DUCE lies in the way its two curricula are tightly integrated in the MARL setting and co-adapt over training.** The two curricula cooperate: early training is dominated by prior-guided samples and therefore benefits from conservative optimization; as prior guidance shrinks and the online policy improves, the sample distribution shifts toward online exploration, the regularization weight decreases, and policy updates become progressively more exploratory.
>
>    More concretely, under the task-difficulty curriculum, we introduce $p_{\text{guide}}$, which determines at the beginning of each episode whether agents follow pure online exploration or a prior-guided curriculum. This mechanism increases the diversity of early collected samples and mitigates the overly restrictive penalties imposed by the conservative regularizer in early training. As shown in Fig. 4, on the 6h_vs_8z medium-replay task, DUCE remains robust to the quality of the prior policy and still yields substantial gains even when the prior’s average win rate is as low as 0.05. Furthermore, Section 6.5 demonstrates that DUCE is compatible with diverse types of prior policies, such as **LLM-generated rule-based scripts** and **existing neural models**—both of which are much easier to obtain in practice—thereby significantly expanding the practical applicability of DUCE.
>
>    We have added a description of the SC-MAIRL algorithm in the Section 2 (Related Work).

---

> > ### Author Response · Authors · 2025-11-28
> >
> > Dear Reviewer wwsc,
> >
> > As the author–reviewer discussion period will conclude soon, we would be grateful if you could kindly take a moment to look over our responses to your comments. If you have any additional questions or feedback, we would be very happy to address them while the discussion window is still open.
> >
> > Thank you very much for your time and consideration.
> >
> > Best regards, The Authors

---

> ### Author Response · Authors · 2025-11-25
> **Response to Reviewer wwsc (3/7)**
>
> > **W3:** Section 6.3 claims for p_guide that "When the values exceed 0.7,  substantial performance fluctuations emerge during the early training  phase." However, in Figure 6, the blue line for p_guide=1 is at all  stages of training the best or close-to-best performing variant. Since  p_guide=1 is from my understanding the base setting for JSRL, I do not  see where it is shown that this contribution is beneficial. Furthermore, the JSRL paper also contains an ablation called "JSRL-Random  Switching", where the guidance length is randomly varied and not  monotonically decreasing, which conceptually sound similar to the  p_guide introduced here.
>
> **Response to W3:**
>
> 1. **Analyze the benefits of introducing $p_{\text{guide}}$ and supplement the experiments with the setting $p_{\text{guide}} = 1$.**
>
>    We first clarify what we mean by “fluctuation.” In Fig. 6, $p_{\text{guide}} = 1$ corresponds to the standard JSRL setting. It exhibits very rapid early improvement but is followed by a sharp performance drop and large oscillations (particularly around 0.3M–0.5M steps). In contrast, settings with $p_{\text{guide}} < 1$ show slightly slower initial improvement but the learning curves are more stable in the early stage and achieve slightly higher mid- to late-stage performance.
>
>    To further illustrate the effect of introducing $p_{\text{guide}} $, we conducted additional experiments on Corridor and 5m_vs_6m under both medium and medium-replay settings in Appendix A.5.7. Specifically, we designed a variant DUCE-no $p_{\text{guide}} $, which is identical to DUCE except that we fix $p_{\text{guide}} = 1$. As shown in Fig. 19 of Appendix A.5.7, **DUCE outperforms DUCE-no $p_{\text{guide}} $**, especially in the early stage of training, where DUCE achieves faster performance improvement. This indicates that the setting of  $p_{\text{guide}} = 1$ hampers MARL training in these tasks. The main reason is that $p_{\text{guide}} = 1$ causes the early replay buffer to contain an excessively high proportion of prior-guided samples. While this initially benefits conservative updates, the offline regularizer penalizes out-of-distribution Q-values so strongly that exploration becomes restricted, which in turn slows down performance improvement. By contrast, introducing $p_{\text{guide}}$ means that, at the beginning of each episode, we decide whether agents follow pure online exploration or a prior-guided curriculum. This simple mechanism increases the diversity of early collected samples and mitigates the overly restrictive penalties imposed by the conservative regularizer in early training, thereby leading to faster and more stable learning.
>
> 2. **New comparison to JSRL-Random Switching**
>
>    To directly address the reviewer’s suggestion, we added a comparative experiment with JSRL-Random Switching in Appendix A.5.6. In this experiment, we design **DUCE-random**, which mimics the guidance scheme of JSRL-Random Switching by randomly sampling the number of guide steps in each episode while keeping $p_{\text{guide}} = 1$. As shown in Fig. 18, results on 6h_vs_8z and 5m_vs_6m (both medium and medium-replay) show that DUCE-random performs substantially worse than DUCE. The main reason is that although Random Switching increases the diversity of collected samples by randomly varying the guidance length and thereby stabilizes early training, it does not specify a condition under which prior guidance is terminated. As a result, the prior policy continues to dominate even when the MARL agent has surpassed it in the mid-to-late stages of training, which harms overall performance.
>
> 3. **Conceptual difference between JSRL-Random Switching and $p_{\text{guide}}$**
>
>    Conceptually, these two schemes are distinct. JSRL-Random Switching (and its DUCE-RandomSwitch variant) always follows the guidance policy during the guided segment of an episode; randomness only affects the duration of guidance. In contrast, our $p_{\text{guide}} < 1$ scheme randomizes *whether* an entire episode is guided or unguided, yielding a mixture of fully guided and unguided episodes. In the presence of an offline regularizer, this distinction is crucial: unguided episodes provide the online policy with room to explore beyond the prior, preventing overly conservative behavior in the early stages of training. We have added a clarification in Appendix A.5.6 to explicitly discuss this conceptual difference between JSRL-Random Switching and the $p_{\text{guide}} < 1$ design.

---

> ### Author Response · Authors · 2025-11-25
> **Response to Reviewer wwsc (4/7)**
>
> > **W4:** From my understanding, the primary difference to the standard CFCQL loss is linearly scheduling the weight alpha ...
>
> **Response to W4:** Our contribution does not lie in the novelty of non-constant regularization weights per se, but rather in (i) **how this schedule is integrated into a dual-track curriculum for MARL cold-start**, and (ii) **the empirical finding that a simple linear decay is both effective and robust across diverse tasks**. To clarify this point, we conducted the following additional analyses:
>
> 1. **Adaptive Behavior Cloning approach does not work well for cold start MARL**
>
>    We have discussed Adaptive BC approach in the related-work section. Adaptive BC adaptively tunes the BC weight during offline-to-online RL to prevent sudden performance drops in single-agent fine-tuning. However, it relies on a relatively complex PID-style controller and requires extensive hyperparameter tuning (the paper explicitly states that grid search is needed, making it cumbersome). Moreover, the ablation in Section 5 (third subsection) shows that adaptive tuning leads to larger fluctuations during training and achieves performance similar to using a fixed weight.
>
>    By contrast, DUCE adopts a simple, monotonically decaying offline-regularization weight, embedded within a dual-track curriculum tailored to MARL. We demonstrate that this lightweight mechanism is sufficient to avoid collapse (as revealed in our preliminary studies in Section 4) and remains robust to a wide range of initial weights and decay lengths across multi-agent benchmarks.
>
> 2. **Evaluation of a variety of decaying schedule**
>
>    In Appendix A.5.8, we provide a systematic evaluation of different decay schedules in the optimization-based curriculum. Our baseline DUCE uses a **linear decay** of the regularizer weight from $\alpha_\text{start}$ to  0 over $T_\alpha$ steps. On top of this, we additionally consider:
>
>    - **DUCE-cosine**, which uses a **cosine decay** schedule
>       $\alpha_t = \frac{1}{2}(\alpha_\text{start})\left(1 + \cos(\pi t / T_\alpha)\right).$
>       This is a **monotone, non-periodic half-cosine** that smoothly interpolates from $\alpha_\text{start}$ to 0: the weight decreases once and does **not** rise again. It also introduces **no additional hyperparameters** beyond $\alpha_\text{start}$, and $T_\alpha$, i.e., the same parameters used by the linear schedule. Cosine decay is a standard choice in DRL for exploration parameters.
>    - **DUCE-fixed**, which keeps the regularizer weight fixed at $\alpha_\text{start}$ throughout training (no decay).
>
>    In all experiments, we use the **same $(\alpha_\text{start}, T_\alpha)$** across the different schedules, without additional tuning for each schedule.  As shown in Fig. 20 of Appendix A.5.8, results on 6h_vs_8z and 5m_vs_6m (medium and medium-replay) show that **DUCE and DUCE-cosine behave very similarly**: both achieve fast and stable learning and reach comparable final performance. In contrast, **DUCE-fixed fails to learn effective policies**, consistent with our Observation in Section 4: while adding an offline RL loss mitigates distribution shift and improves early-stage stability, maintaining a large penalty throughout training restricts exploration, confines collected samples to a narrow region of the state space, and eventually leads to overfitting.
>
>    Taken together, these results suggest that **the key property is simply that the regularizer weight decays over time**; the exact functional form of the decay (linear vs. cosine) is not critical as long as it is monotone. For this reason, and for simplicity and ease of reproduction, we retain the linear decay schedule as the default choice in DUCE.
>
> 3. **DUCE is robust to $\alpha_\text{start}$ and $T_\alpha$**
>
>    The sensitivity analysis regarding the initial weight $\alpha_\text{start}$ and decay period $T_\alpha$ in the optimization-based curriculum is primarily intended to show that DUCE is robust to $\alpha_\text{start}$ and $T_\alpha$. As shown in Fig. 8, varying the initial weight $\alpha_\text{start}$ from 1 to 10 affects only the stability in the early phase but has minimal impact on overall training efficiency, demonstrating DUCE’s strong robustness to this parameter. Similarly, when $T_\alpha$ varies from 0.1M to 1M, overly small or overly large decay periods reduce early efficiency, but mid- and late-stage performance recovers quickly. The overall impact is negligible, again confirming robustness. DUCE’s insensitivity to these hyperparameters shows that the method is practical and does not require complex tuning.
>
>    Therefore, the claim that limited influence from the initial weight and decay period diminishes the contribution is imprecise. Our sensitivity results demonstrate that DUCE effectively addresses the cold-start MARL problem under prior guidance **while remaining robust and easy to tune**, which further strengthens the practical significance of the method.

---

> ### Author Response · Authors · 2025-11-25
> **Response to Reviewer wwsc (5/7)**
>
> > **W5:** In general, the evaluation is performed on different scenarios within  the SMAC environment. However, I consider this somewhat limited. In  particular, the authors only evaluate on discrete action spaces.  Previous work such as CFCQL has used SMAC, MPE and MA-MuJoCo in their  evaluations.
>
> **Response to W5:** We have added experiments in Appendix A.5.3 evaluating DUCE on the widely recognized **MA-MuJoCo** benchmark to demonstrate its applicability to continuous-action multi-agent settings.
>
> 1. **Baselines.**
>    Following our findings on SMAC v1/v2, where ft-QMIX already served as a challenging baseline, we now move to continuous-action MARL settings. Since ft-QMIX cannot be directly applied to continuous action spaces, we select **MATD3**, a widely used actor–critic method for continuous cooperative MARL, as a strong baseline, together with **OVMSE**. Because OVMSE has no official implementation, we re-implemented OVM and SE based on the MATD3 code in the **HARL** repository; all other implementation details follow Section 6.1. For fairness, DUCE uses the *same prior policy* as that produced by OVMSE during its offline stage. To adapt DUCE to continuous action spaces, we adopt **MATD3** as the underlying MARL algorithm, using its implementation from HARL.
>
> 2. **Experimental setup.**
>    We additionally conducted experiments on the widely recognized **Multi-agent MuJoCo (MA-MuJoCo)** benchmark, including challenging scenarios: *HalfCheetah-v2 3×2*, *HalfCheetah-v2 2×3* and *HalfCheetah-v2 6×1*. Each scenario has two dataset tasks—*medium* and *medium-replay*.
> Following the procedure described in the CFCQL paper, datasets were constructed using the **MATD3** algorithm: the medium datasets contain 1000 trajectories, while the medium-replay datasets contain no more than 1000 trajectories (full details in Table 3 and  Appendix A.5.3).
>
> As shown in Fig. 15 of Appendix A.5.3, **DUCE markedly outperforms MATD3 and OVMSE**, converging faster and achieving higher asymptotic performance, even though OVMSE benefits from warm starts that initially give it an advantage.
>
> The HARL repository used in our implementation is available at: [github](https://github.com/PKU-MARL/HARL)
>
> ### Regarding main contributions
>
> > **W6:** In section 3, the definition of domain of pi is given as o_t x a_t. However, o_t and a_t are not defined.
>
> **Response to W6:** We apologize for the notational ambiguity. In Section 3, we have added explicit definitions: $o_t$ denotes the observation at time $t$ , and $a_t$ denotes the action of an agent at time $t$.
>
> > **W7:** Observation II claims that Fig. 2c shows Q_tot of JSRL-CFCQL to "exhibit sharp early fluctuations before stabilizing", but JSRL-QMIX has much  larger shaded area whereas JSRL-CFCQL is as narrow as ft-QMIX?
>
> **Response to W7:** Thank you for pointing out the confusing phrasing. Our intention was to describe that in JSRL-CFCQL, $Q_{\text{tot}}$ rises rapidly and then remains at an elevated level while exhibiting large TD errors. Based on both the magnitude of $Q_{\text{tot}}$ and the TD errors, JSRL-CFCQL suffers from even more severe overestimation than JSRL-QMIX. We have revised the description of $Q_{\text{tot}}$ for JSRL-CFCQL in Section 4 (Observations).

---

> ### Author Response · Authors · 2025-11-25
> **Response to Reviewer wwsc (6/7)**
>
> > **W8:** All regarding Equation (1), which is the core loss introduced by the  authors: It is not explained what a\_-i notation represents. From CFCQL I expect  this to be the joint action without the action of agent i, but this  should be clearly expressed. $E_{a\sim\eta}[Q(s,a)]$, a should likely be bold because I have interpreted it to be a joint action. The notation $E_{\pi_{i}}(\pi_{i}(s)/\eta_{i}(s))$ is unclear. My best guess is $E_{a\sim\pi_t(\cdot|s)}(\pi_i(a|s)/\eta_i(a|s))$ . If this is correct, there are several issues which I do not see adressed: First of all, since samples are drawn from $\pi$, but division happens through probability of $\eta$, if any sample is never selected by $\eta$, this expected value is not defined. Secondly, without further tricks, I would expect that one must be able to evaluate $\eta(a|s)$. Being able to calculate $\eta(a|s)$ is a stronger requirement than being able to sample $a\sim\eta(\cdot|s)$ , the general requirement for a (black box) stochastic policy. Yet, the  paper claims that it addresses "prior policies without assumptions on  their form or structure". Maybe this statement should be relaxed, or the authors should clarify why this is not a concern. I am aware the CFCQL  mitigates this by training a VAE to mimic the behavior policy, but  whatever is done by the authors should be explicitly stated.
>
> **Response to W8:** We thank the reviewer for these detailed questions; they revealed that our notation and implementation details were not sufficiently explained.
>
> 1. **Clarifying notation**
>
>    $a_{-i}$ denotes the joint action of all agents except agent $i$. In $\mathbb{E}_{\mathbf{a}\sim\eta}[Q(s,\mathbf{a})]$, the bold symbol $\mathbf{a}$ indicates the full joint action.
>
> 2. **Clarifying the computation $\lambda_{i}(s)$ in Eq. 1**
>
>    $E_{\pi_i}(\pi_i(s)/\eta_i(s))$ used as part of the computation of $\lambda_{i}(s)$ in Eq. 1, follows the formulation in the CFCQL paper. In discrete action spaces, $\pi(s)$ can be estimated by $\exp\left(\mathbb{E}_{\boldsymbol{a}^{-\boldsymbol{i}}\boldsymbol{\sim}\boldsymbol{\eta}^{-\boldsymbol{i}}}Q(s,a^{\boldsymbol{i}},\boldsymbol{a}^{-\boldsymbol{i}})\right)$ , and we use behavior cloning to train a parameterized $\eta(s)$ from the dataset. In continuous action spaces, $\pi(s)$ is parameterized by each agent’s local policy, and for $\beta_i$, we can use the method of explicit estimation of behavior density in Wu et al. [1], which is modified from a VAE [2] estimator. For the full details of these computations, we refer readers to Appendix B.2 of the CFCQL paper.
>
> The prior policy used for guidance in our method is entirely unrelated to the behavior policy $\eta$. We have added a clarification in the text to explicitly state that the behavior policy and the prior policy have no connection.
>
> [1] Wu, J., Wu, H., Qiu, Z., Wang, J., and Long, M. Supported policy optimization for offline reinforcement learning. In Advances in Neural Information Processing Systems, 2022.
>
> [2] Kingma, D. P. and Welling, M. Auto-encoding variational bayes. arXiv preprint arXiv:1312.6114, 2013.

---

> ### Author Response · Authors · 2025-11-25
> **Response to Reviewer wwsc (7/7)**
>
> ## Questions
>
> > **Q1:**  I would ask the authors to clarify my questions also mentioned in  Weaknesses regarding handling behavior policies for which a probability  density function is not straightforward to calculate. I would appreciate if they could further clarify the points regarding notation.
>
> **Response to Q1:** We have clarified the notational issues in our response to W8 and explained why computing action probabilities for the behavior policy is not required.
>
> ### Suggestions regarding current weaknesses
>
> > **Q2:**  I recommend repeating the main experiments with higher step number such that performance of all methods saturates (or deteriorates if this happens) or removing the claim regarding better asymptotic performance.
>
> **Response to Q2:** Following your suggestion, we have added experiments with a 10M training budget and provided a detailed explanation in our Response to W1.
>
> > **Q3:**  I suggest citing the work mentioned in Weaknesses regarding JSRL in a MARL context and discussing the differences to this work.
>
> **Response to Q3:** We have cited the work you mentioned and provided a detailed comparative analysis, as discussed in our Response to W2.
>
> > **Q4:**  It would be interesting to further analyze how similar the previously  suggested JSRL-Random Switching curriculum is to the proposed  p_guide<1, since both introduce random ordering and in general extend the analysis of this contribution.
>
> **Response to Q4:** We have extended the comparison between $p_{\text{guide}} < 1$ and JSRL-Random Switching. Further details are provided in our Response to W3.
>
> > **Q5:**  Similarly, I would recommend extending the analysis on different  schedules for the regularization loss, especially compared to a variety  of constant factors, and clearly showing the benefit on a variety of  tasks as this is a claimed contribution.
>
> **Response to Q5:** We have extended the analysis of different offline regularization schedules across multiple tasks. Details are provided in our Response to W4.
>
> > **Q6:**  Given the broad claims of applicability, especially that the proposed  framework is defined on arbitrary action spaces and not just discrete  ones, I would suggest further analysis to its benefits on continuous  action spaces, or since this may be difficult within the time frame,  clarify the limitations in evaluation.
>
> **Response to Q6:** We have added experiments on the widely recognized continuous-action MA-MuJoCo benchmark; details are provided in our Response to W5.

---

### Official Review · Reviewer_cXRA · 2025-10-25

**Soundness:** 4
**Presentation:** 4
**Contribution:** 3
**Rating:** 6
**Confidence:** 3

**Summary:**

This paper proposes an offline-to-online MARL approach, involving two types of learning curricula run in parallel: one external, changing the difficulty of tasks as represented by the length of required planning by the learned policy; the other, by gradually shifting the training loss from offline-RL loss to purely online loss.

The paper demonstrates that the combination of these two strategies produces policies which outperform strong baselines on challenging multi-agent coordination tasks.

**Strengths:**

As far as I am aware, this paper is indeed an initial study into cold-start offline MARL algorithms, and the contributions of experimenting with diverse sources of prior policies are indeed novel.

The contribution of the paper is well motivated and justified. In particular, I liked section 4, where the authors present qualitative results supporting the motivation for their proposed method.

Experiments are comprehensive and include reasonable ablations. Results on the challenging SMAC tasks seem to outperform strong baselines.

**Weaknesses:**

Some minor concerns and questions arise when reading the paper.

Line 309: “To strengthen these warm-start baselines, their online training is allowed to leverage offline data, with 30% of training samples drawn from the offline buffer”: Does this actually strengthen the baseline, or rather induce detrimental performance since this data is from a different distribution to the policy? It might be helpful to consider techniques for including offline prior data in RL training for this type of baseline (eg. RLPD).

Nit picks:

- Line 348: text seems cut (missing start of sentence)
- A slight concern on the side of novelty is that the scheduled weighting of the QMIX and CFCQL losses should probably be considered a scheduled hyperparameter adjustment, rather than a curriculum.

Other concerns follow in the questions section below. If these concerns and questions are answered adequately during the rebuttal period, I am happy to raise my score accordingly.

**Questions:**

1. Section 3 (preliminaries) - as suggested by the reward definition, and the definitions of Q functions in later sections - does the algorithm proposed in this paper only deal with reward shared among all agents (i.e. cooperative scenarios only)?
2. What are the prior policies used for the variants of JSRL?
3. Why are only sub-optimal policies considered? Why not work with expert policies, if they are available?
4. In eq. 1, is the Q function in the CFCQL loss a single-agent Q function or the total Q function for all agents, similar to $Q_{tot}$? If it’s the former, doesn’t the mixing between these two losses cause issues with scaling of gradients and stability of training?
5. Are the prior policies used for guidance in the external curriculum identical to the ones used as behavior policies for the CFCQL loss?

---

> ### Author Response · Authors · 2025-11-25
> **Response to Reviewer cXRA (1/4)**
>
> We sincerely thank the reviewer for the careful reading, constructive questions.
>
> Below we respond to each point raised by the reviewer and clarify the corresponding changes we will make in the revised version.
>
> ## Weaknesses
>
> > **W1:** The comment questions whether sampling 30% of training data from the offline buffer genuinely strengthens warm-start baselines, since the offline data may be distributionally different from the online policy. It also suggests considering established techniques—such as RLPD—for incorporating offline prior data into RL training.
>
> **Response to W1:**  Our intention is precisely to **strengthen the warm-start baselines** and avoid giving DUCE any unfair advantage.
>
> As pointed out by the authors of **RLPD** in their ICLR 2025 paper[1], the key finding is:
>
> “*Training only on online experience during fine-tuning without offline data retention can destroy how well the model fits offline data: despite attaining comparable TD-errors on the online data to the setting when offline data is retained, TD-errors under the offline distribution grow larger.*”
>
> Concretely, in the warm-start setting, the baselines already begin from policies trained on the offline dataset. For these baselines, reusing the **same offline buffer** during online fine-tuning is a *natural extension* of the offline-to-online workflow:
>
> - The offline trajectories were generated by their **own prior policies**, so the distributional mismatch is mild.
>
> - The offline samples function as **additional replay** for rewarding transitions, which is standard practice in O2O RL.
>
> - This avoids catastrophic forgetting of behaviors learned during offline training.
>
> To further clarify this issue, we added an additional experiment in Appendix A.5.2 in which **offline data is entirely removed during online training**. Specifically, OVMSE, InSPO, CFCQL, and MACal-QL are trained **without using any offline data** once online learning begins. Fig. 14a reports the performance of these warm-start baselines *without* offline data and Fig. 14b reports their performance *with* offline data—on *6h_vs_8z* (medium and medium-replay). As shown in Fig. 14a,  **DUCE significantly outperforms OVMSE, InSPO, CFCQL, MACal-QL, and ft-QMIX**. Comparing Fig. 14a and 14b, we observe that removing offline data during fine-tuning causes a **substantial performance drop** for CFCQL and MACal-QL, leaves InSPO largely unchanged, and leads to a **notable degradation** for OVMSE on *6h_vs_8z-medium*. These results strongly support the conclusion that training solely on online experience during fine-tuning, without retaining offline data, can severely impair the model’s ability to preserve what it previously learned from the offline data, thereby causing a performance degradation. We have strengthened the explanation of this design choice in Sec. 6.1.
>
> [1] Zhiyuan Zhou, Andy Peng, Qiyang Li, Sergey Levine, and Aviral Kumar. *Efficient online reinforcement learning fine-tuning need not retain offline data.* In **ICLR 2025**, pp. 32343–32368.
>
> > **W2:** Line 348: text seems cut (missing start of sentence)
>
> **Response to W2:**  Thank you for catching this copy-editing error. The sentence was indeed accidentally truncated during formatting and should explicitly include the name of the offline datasets. We have corrected and completed this description in Sec. 6.1.

---

> > ### Author Response · Authors · 2025-11-27
> >
> > Dear Reviewer cXRA,
> >
> > As the author–reviewer discussion period will conclude soon, we would be grateful if you could kindly take a moment to look over our responses to your comments. If you have any additional questions or feedback, we would be very happy to address them while the discussion window is still open.
> >
> > Thank you very much for your time and consideration.
> >
> > Best regards,
> > The Authors

---

> ### Author Response · Authors · 2025-11-25
> **Response to Reviewer cXRA (2/4)**
>
> > **W3:** A slight concern on the side of novelty is that the scheduled weighting of the QMIX and CFCQL losses should probably be considered a scheduled hyperparameter adjustment, rather than a curriculum.
>
> **Response to W3:**  DUCE’s policy-optimization curriculum does adjust the weight of the regularization term in form, but conceptually it is **co-designed** with the task-based curriculum to control how learning transitions from conservative updates to autonomous exploration.
>
> 1. **Progression of optimization regimes.**
>
>    In DUCE, the agent starts in a regime dominated by the conservative offline loss $L_{\text{CFCQL}}$, which biases learning toward exploiting prior-policy data to mitigate distribution shift. As $\alpha(t)$ decays, $L_{\text{CFCQL}}$ gradually relinquishes control and the optimization transitions to a regime dominated by the online ft-QMIX loss, which promotes exploration on newly collected data. This is a structured progression from “offline-like” to “online-like” optimization, not just a cosmetic change of a scalar.
>
> 2. **Complementarity with the task-based curriculum.**
>
>    The externally configured **task-difficulty curriculum** determines **how the prior and online policies interact with the environment**, thereby shaping the state distribution of the collected data as the curriculum progresses. As the guidance horizon shortens, more of each episode is controlled by the online policy, and the data distribution gradually shifts from prior-dominated to exploration-dominated.
>
>    The internally evolved **policy-optimization curriculum** then decides **how to best leverage this curriculum-generated data** for efficient policy updates. It operates on the loss function, gradually shifting updates from conservative to exploratory: early, prior-dominated data is handled with stronger conservative regularization, while later, online-exploration data is allowed to drive increasingly exploratory updates.
>
>    **The core novelty of DUCE lies in how these two curricula are tightly integrated in the MARL setting and co-adapt over training through the replay buffer.** Under the task-difficulty curriculum, the mixed policy (prior + online) interacts with the environment and populates the replay buffer with relatively high-quality trajectories containing a large proportion of prior-guided steps. The policy-optimization curriculum applies conservative updates on this mixed-distribution data to rapidly improve the online policy despite distributional shift. As the online policy improves and the task-difficulty curriculum reduces prior guidance, the fraction of autonomously explored steps in the replay buffer increases; correspondingly, the regularization weight is annealed down, and policy updates become progressively more exploratory. In short, the two curricula cooperate: early training is dominated by prior-guided samples and thus benefits from conservative optimization, whereas later training is driven by online exploration with lighter regularization.
>
> 3. **Empirical evidence that the progression matters.**
>
>    Sec. 6.4 (Fig. 7) shows that: (i) removing the CFCQL offline regularizer entirely (DUCE-noOfflineLoss), and (ii) removing the ft-QMIX TD component while keeping the offline loss weight fixed (DUCE-noOnlineLoss), both lead to severe degradation on challenging tasks.
>
>    This demonstrates that the *temporal progression* of the optimization objective is crucial for successfully leveraging prior guidance; a fixed-weight regularizer is insufficient.
>
>    We have strengthened the description of the innovation of the policy-optimization curriculum in Sec. 5.

---

> > ### Author Response · Authors · 2025-11-28
> >
> > Dear Reviewer cXRA,
> >
> > As the author–reviewer discussion period will conclude soon, we would be grateful if you could kindly take a moment to look over our responses to your comments. If you have any additional questions or feedback, we would be very happy to address them while the discussion window is still open.
> >
> > Thank you very much for your time and consideration.
> >
> > Best regards, The Authors

---

> ### Author Response · Authors · 2025-11-25
> **Response to Reviewer cXRA (3/4)**
>
> ## Questions
>
> > **Q1:** Section 3 (preliminaries) - as suggested by the reward definition, and  the definitions of Q functions in later sections - does the algorithm  proposed in this paper only deal with reward shared among all agents  (i.e. cooperative scenarios only)
>
> **Response to Q1:** In this paper, we primarily focus on cooperative MARL, and we think that DUCE also applies to other CTDE settings, such as mixed cooperative–competitive and fully competitive scenarios.
>
> Concretely:
>
> 1. **Semi-competitive / semi-cooperative settings.**
>    Such environments can be modeled as **general-sum Markov games**, where each agent’s reward contains both a team component and an individual component. CTDE-based MARL algorithms can solve general-sum Markov games: during centralized training, the centralized critic integrates all reward signals—team rewards contribute cooperative gradients, while individual rewards contribute differentiated policy gradients—allowing agents to learn strategies that mix cooperation with competition.
>
> 2. **Fully competitive settings.**
>    Purely competitive tasks can be modeled as **zero-sum Markov games**, where the rewards of two agents $A$ and $B$ are exact negatives of each other. CTDE-based MARL methods can also solve zero-sum games: during centralized training, the centralized critic learns the Q-function for agent $A$; the Q-function for agent $B$ is simply its negative. When updating the actor for agent $A$, we maximize $A$’s Q-value, while when updating agent $B$, we maximize the negative of $A$’s Q-value, forming a distributed adversarial policy-learning process.
>
> Thus, because DUCE sits atop a CTDE MARL backbone, it naturally extends to **general-sum** and **zero-sum** Markov games, and is therefore not limited to cooperative shared-reward environments. We will evaluate DUCE in these two types of scenarios in future experiments.
>
> > **Q2:** What are the prior policies used for the variants of JSRL?
>
> **Response to Q2:** For both JSRL-QMIX and JSRL-CFCQL, we use the same prior policies as in Sec. 6 (Experiments). For example, On the 6h_vs_8z medium-replay task, the prior policy is obtained by training CFCQL on the medium-replay offline dataset, achieving an average win rate of 5% and an average return of 13.33. The performance of prior policies for other tasks is reported in Table 4. We have added this clarification to Sec. 4.
>
> > **Q3:** Why are only sub-optimal policies considered? Why not work with expert policies, if they are available?
>
> **Response to Q3:** A more accurate description is that **our method is not strongly dependent on the quality of the prior policy**. In realistic cold-start scenarios, one typically has access to *feasible but non-expert* priors—such as heuristics, LLM-generated rules, or early-stage learned policies—while near-optimal expert policies are often expensive or unavailable. For this reason, in Sec. 5 we formalize our setting using **suboptimal priors**, defined as policies that clearly outperform random behavior yet remain far from optimal performance. DUCE is designed to convert such suboptimal priors into effective cold-start guidance while avoiding overfitting and distributional shift.
>
> This does **not** mean that expert priors cannot be used. Stronger (expert-level) priors enable DUCE to achieve even higher training efficiency. In Appendix 5.4, we added experiments using expert priors. Specifically, **DUCE-expert** denotes DUCE equipped with an expert policy as the prior. As shown in Fig. 16 of Appendix 5.4, on 5m_vs_6m and 6h_vs_8z (medium and medium-replay), DUCE-expert shows higher training efficiency than DUCE. We have strengthened the discussion in Sec. 5 to clarify that DUCE is fully compatible with expert policies as well.

---

> ### Author Response · Authors · 2025-11-25
> **Response to Reviewer cXRA (4/4)**
>
> > **Q4:** In eq. 1, is the $Q$ function in the CFCQL loss a single-agent $Q$ function or the total $Q$ function for all agents, similar to $Q_{tot}$? If it’s the former, doesn’t the mixing between these two losses cause  issues with scaling of gradients and stability of training?
>
> **Response to Q4:** We apologize for the ambiguity in notation. In our implementation, the $Q$ in the CFCQL loss is the **joint** action-value function $Q_{\text{tot}}$, consistent with the QMIX loss. That is, we do **not** mix per-agent Q-functions with the joint Q-function; both $L_{QMIX}$ and $L_{CFCQL}$ are defined on the same $Q_{\text{tot}}$.
>
> We will explicitly reflect this in Eq. (1) and in the surrounding text to avoid confusion. This design keeps the two losses on a compatible scale and did not cause any stability issues in our experiments.
>
> > **Q5:** Are the prior policies used for guidance in the external curriculum  identical to the ones used as behavior policies for the CFCQL loss?
>
> **Response to Q5:**  In general, they are **not required to be identical**.
>
> - The **prior policies used for external curriculum guidance** are the policies that directly interact with the environment during the prior-guided segments of an episode (e.g., rule-based policies derived from large language models). DUCE is explicitly designed to accommodate priors with different architecture.
>
> - The **behaviour policy $\eta$ in CFCQL** · represents the policy under which the offline dataset was generated. When the prior is  a neural policy，a white-box rule or an LLM-based controller, we still treat its interaction trajectories as defining an empirical behaviour distribution, but we do not require parameter sharing between $\eta$ and the prior used for guidance.
>
> This decoupling is important for the **cold-start** setting we consider: DUCE should be able to benefit from different prior sources without assuming that their network architectures can be reused inside the offline loss. We have added the corresponding clarification in Sec. 5.
>
> We hope that these clarifications and the corresponding manuscript changes address your concerns regarding the baselines, the nature of the optimization-based curriculum, the scope of applicability, and the technical details of the loss design.

---

### Official Review · Reviewer_wwvQ · 2025-10-28

**Soundness:** 2
**Presentation:** 2
**Contribution:** 2
**Rating:** 2
**Confidence:** 3

**Summary:**

This work introduces a framework for training multi-agent RL problems when making use of an informative prior policy. This prior is not necessarily optimal, but may provide structured guidance that assists in exploration. The proposed method, DUCE, does not use warm-start techniques and instead initializes from a random policy. DUCE operates by building off the jump-start RL framework, and starts episodes by first following the trajectory defined by a prior guidance policy. A curriculum is established where the guidance policy is replaced by the live policy at increasingly early intervals. At the same time, an "offline RL" auxiliary loss is computed on a fixed dataset, and the weighting of this loss is decreased over the training interval. Experiments are conduced over the SMAC domain, and DUCE is shown to outperform the main baselines.

**Strengths:**

The ideas presented in this paper are justified and empirically show an improvement. The paper provides numerical analysis on the pitfalls of naive baselines, including reproducing the "warm-start collapse" phenomenon where warm start policies show an initial drop in performance. Over the standard benchmarks, the proposed method shows a consistent improvement. Ablations are presented on the two core proposals (prior guidance and offline RL).

**Weaknesses:**

- The proposed method introduced significant complexity. In general it is undesirable to introduce new hyperparameters, and the proposed method can be seen as a "bag of tricks" combining JSRL and the offline CFCQL, along with a tuned curriculum that modulates their effects over training. The resulting method does not appear immediately practical as it introduces additional complexity that must be tuned.
- The paper's novelty is not immediately apparent. The two adjustments presented have been detailed in previous works, and the paper does not present any concrete analysis on the *cross-interaction* of these two adjustments.
- The loss function in equation 1 combines two styles of TD loss. This could be strengthened by an ablation to just use the same TD loss (but over an offline vs. online state distribution).
- It is unclear how much of the insights in this paper are specific to multi-agent RL, vs. offline-to-online RL methodology in general. The proposed algorithm does not include any specific multi-agent components. To strengthen the multi-agent specificity, it may be beneficial to highlight performance on single-agent tasks as well, and/or show explicit analysis that specific phenomena only appear in the multi-agent setting.

**Questions:**

- Do the curves in Fig 1 come from real data? I would be worried that the trends presented in the stylized graph are not neccessarily reflected in a real experiment, or are domain dependent.
- How much does the gain from DUCE depend on the specific prior policy?
- On page 5, it is stated that h_0 is set to the average steps required by the prior policy to solve the task. However, it is later states that the prior policies are designed to be suboptimal and have a < 100% success rate. In that case, how is h0 selected?
- In equation 1, there are three Q networks referenced -- Q, Q_tot, and Qbar. Qbar is defined as the target network, but what is the difference between Qtot and Q?
- In equation 1, how is pi(s) calculated, when pi is originally defined as pi(a|s)?
- Is there a way to measure the "diversity" of states visited by the guiding policy vs. the cold start policy? This form of analysis may improve the paper to describe concrete phenomena rather than potentially domain-specific or black-box proposed improvements.

Minor:
- It is unclear/undefined what "guidance step size" in the top of page 5 means.
- The paper would be strengthened by an explicit explanation of what JSRL and QMIX do.
- An explicit algorithm box would improve clarity, including which hyperparameters need to be tuned per-task.

---

> ### Author Response · Authors · 2025-11-25
> **Response to Reviewer wwvQ (1/5)**
>
> We thank the reviewer for the careful reading and the constructive feedback.
>
> Below we address each concern in detail and describe the corresponding changes made to the manuscript.
>
> ## Weaknesses
>
> > **W1:** The proposed method introduced significant complexity. In general it is  undesirable to introduce new hyperparameters, and the proposed method  can be seen as a "bag of tricks" combining JSRL and the offline CFCQL,  along with a tuned curriculum that modulates their effects over  training. The resulting method does not appear immediately practical as  it introduces additional complexity that must be tuned.
>
> **Response to W1:** We appreciate this concern and concur that practicality and manageable complexity are crucial for MARL methods.
>
> DUCE’s dual-track design introduces only two new hyperparameters: the interleaving probability $p_{\text{guide}}$ and the decay period $T_{\alpha}$ of the offline loss. Conceptually, $p_{\text{guide}}$ controls how often the agents follow prior-guided episodes versus purely online exploration, while $T_{\alpha}$ controls the  decay period of conservative offline loss during training. Our ablations in Sec. 6.3 and 6.4 show that, as long as these two hyperparameters stay within a reasonably wide operating range around their default values, the performance curves change only mildly; noticeable degradation occurs only when they are pushed to extreme values (e.g., $p_{\text{guide}}$ very close to 0, or $T_{\alpha}$ extremely large or extremely small). We recommended $p_{\text{guide}} = 0.5$ and $T_{\alpha} = 100\text{K}$ as default values, allowing readers to adopt them directly without further tuning.
>
> We also inherit several parameters from prior work: the number of curriculum stages $m$ and the performance threshold $\beta$ from JSRL, and the initial offline-regularization weight $a_{\text{start}}$ from CFCQL. These hyperparameters are all easy to tune. The JSRL paper notes that $m$ only needs to be large enough to allow smooth curriculum progression; in our experiments, we typically set $m = 10$. Both $\beta$ and $a_{\text{start}}$ have clear recommended ranges in their respective papers. We adopt $\beta = 0.95$ and $a_{\text{start}} = 1$ as default values, and the sensitivity analyses in Sec. 6.3 and Sec. 6.4 show that only minor performance variation occurs without modifying them.
>
> In contrast, recent algorithms in related domains typically introduce two or more tunable hyperparameters, as shown in Table W1, whereas DUCE introduces only two, each with recommended default values. In summary, DUCE does not introduce complex hyperparameter tuning. We have added the corresponding hyperparameter descriptions in Appendix A.3.2.
>
> **Table W1** Hyperparameter configuration of DUCE and recent algorithms.
>
> | Method | Number of hyperparameters | Specific hyperparameters                                     |
> |:--|:---|:--|
> | OVMSE [1] | 4 | initial memory coefficient; final memory coefficient; memory decay period; offline data ratio. |
> | InSPO [2] | 2 | conservatism temperature; entropy temperature. |
> | ComaDICE [3] | 2 | regularization coefficient; scaling factor |
> | CFCQL [4] | 2 | conservatism weight; temperature coefficient. |
> | HASAC [5] | 3 | temperature parameter; automated temperature tuning switch; temperature learning rate |
> | **DUCE** | **2** | $p_{\text{guide}}$ and the decay period $T_\alpha$, with recommended default values. |
>
> [1] Hai Zhong, Xun Wang, Zhuoran Li, and Longbo Huang. Offline-to-online multi-agent reinforcement learning with offline value function memory and sequential exploration. In Proceedings of the 24th International Conference on Autonomous Agents and Multiagent Systems. International Foundation for Autonomous Agents and Multiagent Systems, 2025.
>
> [2] Zongkai Liu, Qian Lin, Chao Yu, Xiawei Wu, Yile Liang, Donghui Li, and Xuetao Ding. Offline multi-agent reinforcement learning via in-sample sequential policy optimization. In Proceedings of the AAAI Conference on Artificial Intelligence, volume 39, pp. 19068–19076, 2025.
>
> [3] Nguyen T H, Mai T A. Comadice: Offline cooperative multi-agent reinforcement learning with stationary distribution shift regularization[C]//The Thirteenth International Conference on Learning Representations. 2024.
>
> [4] Jianzhun Shao, Yun Qu, Chen Chen, Hongchang Zhang, and Xiangyang Ji. Counterfactual conservative q learning for offline multi-agent reinforcement learning. Advances in Neural Information Processing Systems, 36:77290–77312, 2023.
>
> [5] Liu J, Zhong Y, Hu S, et al. Maximum Entropy Heterogeneous-Agent Reinforcement Learning[C]//The Twelfth International Conference on Learning Representations, 2024.

---

> > ### Author Response · Authors · 2025-11-27
> >
> > Dear Reviewer wwvQ,
> >
> > As the author–reviewer discussion period will conclude soon, we would be grateful if you could kindly take a moment to look over our responses to your comments. If you have any additional questions or feedback, we would be very happy to address them while the discussion window is still open.
> >
> > Thank you very much for your time and consideration.
> >
> > Best regards,
> > The Authors

---

> ### Author Response · Authors · 2025-11-25
> **Response to Reviewer wwvQ (2/5)**
>
> > **W2:** The paper's novelty is not immediately apparent. The two adjustments  presented have been detailed in previous works, and the paper does not  present any concrete analysis on the *cross-interaction* of these two adjustments.
>
> **Response to W2:**  Our work is not a simple combination of JSRL and CFCQL. Our work involve multiple novel designs (such as tightly coupled **dual-track curricula** that coordinate (i) a curriculum based on externally configured task difficulty, and (ii) a curriculum based on internally evolved policy optimization) and adapt them to a new problem setting--cold-start MARL with prior-guided exploration.
>
> To disentangle generic benefits of JSRL from MARL-specific challenges (non-stationarity and credit assignment), we first cast the multi-agent task a multi agent problem into a **centralized control problem**, where a single agent observes the global state and outputs the joint action. On top of this centralized formulation, we additionally implement **IQL** and **JSRL-IQL**, i.e., applying JSRL to an IQL-style single-agent learner. The results in Fig. 11 of Appendix A.4.3 show that in the early stage of training, JSRL-IQL exhibits a rapid performance increase compared with IQL, confirming that JSRL’s curriculum indeed provides a clear “jump-start” compared with no curriculum in the centralized setting. This is precisely why we regard JSRL as a promising starting point. However, centralized control suffers from the curse of dimensionality, making it difficult for JSRL-IQL to learn effective policies.
>
> When we migrate this idea **back to the multi-agent setting**, simple combinations of JSRL with standard MARL backbones **fail to deliver the same benefit**. We specifically observe two failure modes:  **(i)** extending JSRL to MARL with **ft-QMIX** (a representative online MARL method)  as the backbone **(JSRL-QMIX**) yields worse training efficiency than ft-QMIX itself and suffers from severe overestimation of $Q_{\text{tot}}$ under prior guidance due to distributional shift (Sec. 4, Observation I); and **(ii)** augmenting JSRL-QMIX with a CFCQL(a representative offline MARL method)-based offline regularizer (**JSRL-CFCQL**) temporarily improves early performance, but its excessive penalties constrain exploration, confine collected samples to a narrow local region of the state space, inflate in-distribution Q-values, and ultimately lead to overfitting (Sec. 4, Observation II).
>
> These negative results show that **simply “plugging JSRL into QMIX or CFCQL” is not sufficient** in MARL. Motivated by these observations, we propose DUCE, a dual-track curriculum tailored to cold-start MARL with prior-guided exploration.  **The core novelty of DUCE lies in the way its two curricula are tightly integrated in the MARL setting and co-adapt over training**. Under the task-difficulty curriculum, the mixed policy formed by the prior and online policies interacts with the environment to generate relatively high-quality samples (with a high proportion of prior-guided data), which are stored in the replay buffer. The policy-optimization curriculum then applies conservative policy updates on this mixed-distribution data to rapidly improve the online policy despite distributional shift. As the online policy improves and the task-difficulty curriculum progresses, the proportion of autonomous exploration samples increases, and these new samples enter the replay buffer. Subsequently, the policy-optimization curriculum gradually shifts from conservative updates to exploration-driven updates, adapting to the evolving sample distribution. In short, the two curricula cooperate: the early stage is dominated by prior-guided samples and thus benefits from conservative optimization; as guidance shrinks and online performance improves, the sample distribution shifts toward online exploration, the regularization weight decreases, and policy updates become progressively more exploratory.
>
> This cross-interaction is summarized in Figure 3 and in the concluding paragraph of Sec. 5. We have revised the method description to highlight how the two curricula jointly address the cold-start MARL problem under prior guidance and why their integration is not an ad hoc combination.
>
> Sections 6.3 and 6.4 include ablation variants that remove one curriculum (DUCE-noGuide and DUCE-noOfflineLoss). These variants remain competitive only on a subset of tasks and perform markedly worse than DUCE on more challenging maps, demonstrating that neither curriculum alone is sufficient. We clarify in Sections 6.3 and 6.4 that the ablations reveal the complementary nature of the two curricula rather than merely showing that each component offers marginal benefit.

---

> > ### Author Response · Authors · 2025-11-28
> >
> > Dear Reviewer wwvQ,
> >
> > As the author–reviewer discussion period will conclude soon, we would be grateful if you could kindly take a moment to look over our responses to your comments. If you have any additional questions or feedback, we would be very happy to address them while the discussion window is still open.
> >
> > Thank you very much for your time and consideration.
> >
> > Best regards, The Authors

---

> ### Author Response · Authors · 2025-11-25
> **Response to Reviewer wwvQ (3/5)**
>
> > **W3:** It is unclear how much of the insights in this paper are specific to  multi-agent RL, vs. offline-to-online RL methodology in general. The  proposed algorithm ...
>
> **Response to W3:**  **First**,  DUCE is *not* an O2O RL method, and the distinction is substantial. O2O RL typically focuses on leveraging a prior policy obtained via offline pre-training to **initialize a single-agent policy**, followed by online fine-tuning. In our work, however, the underlying problem is **cooperative multi-agent RL**: **QMIX**, used as our online backbone, is a standard value-factorization algorithm for cooperative MARL, and **CFCQL** is the *multi-agent* extension of CQL designed for offline MARL under CTDE. In this MARL setting, DUCE does **not** initialize the online policy with the prior, which is a key difference from O2O RL methods; instead, it uses the prior solely to interact with the environment and generate guidance data. This design makes DUCE compatible with diverse forms of prior policies—including, but not limited to, existing neural models and rule-based scripts generated by large language models. Sec. 6.5 further demonstrates that DUCE can exploit LLM-generated rule-based scripts to significantly accelerate training efficiency and improve MARL performance in cold-start scenarios.
>
> **Second**, the insights provided by this work are *intrinsically multi-agent*, even though we deliberately start from a single-agent surrogate to disentangle where the difficulties come from. We first cast the multi-agent task a multi agent problem into a **centralized control problem**. On top of this centralized formulation, we additionally implement **IQL** and **JSRL-IQL**, i.e., applying JSRL to an IQL-style single-agent learner. The results in Fig. 11 show that JSRL’s curriculum indeed provides a  “jump-start” compared with no curriculum in the centralized setting. This justifies using JSRL as a promising starting point. However, both JSRL-IQL and IQL achieve very low win rates because centralized control still suffers from the curse of dimensionality.
>
> When we migrate this idea **back to the genuine multi-agent setting**, simple combinations of JSRL with standard MARL backbones **fail to deliver the same benefit**. We specifically observe two failure modes:  **(i)** Extending JSRL to cooperative MARL with **ft-QMIX**  as the backbone (**JSRL-QMIX**) yields worse training efficiency than ft-QMIX itself and suffers from severe overestimation of $Q_{\text{tot}}$ under prior guidance due to distributional shift (Sec. 4, I);  **(ii)** Augmenting JSRL-QMIX with an offline regularizer based on **CFCQL** (**JSRL-CFCQL**) temporarily improves early performance, but its excessive penalties constrain exploration, confine collected samples to a narrow local region of the state space, inflate in-distribution Q-values, and ultimately lead to overfitting (Sec. 4, II).
>
> These failure modes arise from **multi-agent–specific challenges**—namely, the exponentially large joint action space, non-stationarity, and credit assignment—which amplify distributional shift and value overestimation. They show that naïve prior-guided cold-start adaptations of single-agent ideas (JSRL-QMIX, JSRL-CFCQL) encounter failures that **appear only in MARL** and are not simply inherited from the single-agent setting. Our contribution is therefore not a straightforward transplantation of single-agent prior-guided methods into MARL; rather, we **identify structural failure modes unique to MARL**, which in turn motivate the design of the dual-track DUCE framework.
>
> **Third**, DUCE’s loss design further reflects multi-agent characteristics. The offline regularization term is introduced specifically to address distribution shift and value overestimation exacerbated by the exponentially expanding joint action space. Under the CTDE paradigm, we employ a counterfactual strategy: when computing the regularizer for agent $i$, only this agent’s action is replaced with an OOD action, while the actions of other agents are fixed to those in the dataset. This avoids overly sampling the joint action space. The online loss also incorporates multi-agent–specific components, such as combining local utilities into a joint value via a non-negative mixing network to compute TD errors. In Sec. 4, we emphasize that DUCE is designed precisely to address phenomena that arise *only* in MARL and that its design embodies multi-agent structure throughout.
>
> **Finally**, we supplement Sec.4 and Appendix A.4.2 with further experiments on naive prior-guided cold-start methods (JSRL-QMIX, JSRL-CFCQL) across tasks with varying numbers of agents, including the 5m_vs_6m and 6h_vs_8z medium and medium-replay settings. The results show that as the number of agents increases, JSRL-QMIX and JSRL-CFCQL exhibit even more severe instability and collapse, further validating that the core problem addressed in this paper is fundamentally multi-agent in nature.

---

> ### Author Response · Authors · 2025-11-25
> **Response to Reviewer wwvQ (4/5)**
>
> > **W4:** The loss function in equation 1 combines two styles of TD loss. This  could be strengthened by an ablation to just use the same TD loss (but  over an offline vs. online state distribution).
>
> **Response to W4:**  Thank you for the suggestion. In Sec. 6.4, we originally conducted an ablation study removing the offline regularization term; to further disentangle the effects of different loss components, we additionally include an ablation that removes the online TD loss. Specifically, we introduce **DUCE-noOnlineLoss**, which removes the online TD loss while keeping the offline regularization weight unchanged. As shown in Figure 7, DUCE-noOnlineLoss performs worse than DUCE on both the 6h vs 8z-medium task and the Zerg 5v5-medium replay task, primarily because its excessive penalties constrain exploration, restrict the sampled trajectories to a narrow local region of the state space, and ultimately lead to overfitting.
>
>
> ## Questions
>
> > **Q1:** Do the curves in Fig 1 come from real data? I would be worried that the  trends presented in the stylized graph are not neccessarily reflected in a real experiment, or are domain dependent.
>
> **Response to Q1:**
>
> We thank the reviewer for highlighting this potential source of confusion. Figure 1 is a *stylized schematic*, not an empirical plot. Its design is based on our Sec. 4 preliminary experiments comparing naïve prior-guided cold-start methods with ft-QMIX, as well as the main results in Sec. 6 comparing DUCE, warm-start O2O MARL, and ft-QMIX. Its sole purpose is to provide an intuitive qualitative illustration of these methods. To avoid misunderstanding, we have revised the caption to explicitly state that Figure 1 is a schematic.
>
> > **Q2:** How much does the gain from DUCE depend on the specific prior policy?
>
> **Response to Q2:**
>
> First, DUCE is effective with **suboptimal priors of varying quality**. As shown in Table 4 of  Appendix A.5.11, on *6h_vs_8z-medium* the prior policy achieves an average win rate of 0.25 and an average return of 15.22, whereas on *6h_vs_8z-medium-replay* the prior achieves only 0.05 win rate and a return of 13.33—reflecting the substantial quality gap arising from training on different offline datasets. Nevertheless, as illustrated in Fig. 4 (Sec. 6.2), DUCE consistently outperforms all baselines on both tasks, indicating that even low-quality priors can significantly accelerate learning when used within DUCE.
>
> Second, DUCE is compatible with **priors of different architectural forms**. In Sec. 6.2, DUCE employs an RNN-based neural policy as the prior. In Sec. 6.5, we further evaluate DUCE with two markedly different types of priors: **(i)** a fully connected neural network prior (**DUCE_FC_Model**) and **(ii)** rule-based scripts generated by a large language model (**DUCE_LLM_Rule**). Both variants substantially accelerate training relative to ft-QMIX, even though the standalone LLM-generated rules achieve near-zero win rates on the evaluated maps.
>
> These results collectively demonstrate that DUCE is not overly sensitive to either the **absolute quality** or the **architectural form** of the prior. We have added explicit statements in Sec. 6.5 to highlight this robustness.
>
> > **Q3:**  On page 5, it is stated that $h_0$ is set to the average steps required by the prior policy to solve the task. However, it is later states that the prior policies are designed to be suboptimal and have a $< 100\% $success rate. In that case, how is h0 selected?
>
> **Response to Q3:**  We apologize for the imprecise wording. We estimate $h_0$ as the average episode length obtained by executing the prior policy alone; when the prior policy fails to complete the task due to its suboptimality, we treat unsuccessful episodes as having the maximum episode length. We have updated the description accordingly in the second paragraph of Sec. 5.
>
> > **Q4:** In equation 1, there are three $Q$ networks referenced -- $Q$, $Q_{tot}$, and  $\bar{Q}_{\mathrm{tot}}$. $\bar{Q}_{\mathrm{tot}}$ is defined as the target network, but what is the difference  between $Q_{tot}$ and $Q$ ?
>
> **Response to Q4:**  Thank you for catching this confusing notation. In Eq. (1), $Q$ and $Q_{tot}$  are intended to denote the same joint action-value function, and $\bar{Q}_{\mathrm{tot}}$ its target network.
>
> To avoid confusion, we have unified the notation so that Eq. (1) uses only $Q_{tot}$  and $\bar{Q}_{\mathrm{tot}}$.

---

> ### Author Response · Authors · 2025-11-25
> **Response to Reviewer wwvQ (5/5)**
>
> > **Q5:** In equation 1, how is pi(s) calculated, when pi is originally defined as $\pi(a|s)$?
>
> **Response to Q5:** We agree that our shorthand notation was unclear. The computation of $\pi(s)$ introduced in Eq. 1 follows the formulation in the CFCQL paper. As in CFCQL, $\pi(s)$ can be explicitly evaluated as $\exp\left(\mathbb{E}_{\boldsymbol{a}^{-i}\boldsymbol{\sim}\boldsymbol{\beta}^{-i}}Q(s,a^i,\boldsymbol{a}^{-\boldsymbol{i}})\right)$ in discrete action spaces, while in continuous action spaces $\pi(s)$ is parameterized by each agent’s local policy. For a more detailed explanation of how $\pi(s)$ is computed, please refer to Appendix B.2 of the CFCQL paper.
>
> > **Q6:** Is there a way to measure the "diversity" of states visited by the  guiding policy vs. the cold start policy? This form of analysis may  improve the paper to describe concrete phenomena rather than potentially domain-specific or black-box proposed improvements.
>
> **Response to Q6:** We have added a qualitative visualization of the state distribution in Appendix A.5.10. Specifically, we apply t-SNE to embed the global states into a three-dimensional space and plot the states visited by the prior policy and the online policy at the beginning and at the end of training. As shown in Fig. 22 of Appendix A.5.10, before training, the state distributions visited by the guiding policy and the cold-start policy differ; comparing the before-and-after results, we observe that the the cold-start policy’s distribution gradually moves across the embedding space toward and through the region occupied by the guiding policy, and eventually expands into a broader area beyond the guiding policy’s coverage.
> This evolution indicates that DUCE’s dual-track curriculum uses the guiding policy as a directional guide for exploration—steering the cold-start policy toward informative regions of the state space while allowing it to further explore beyond the guiding policy’s support, thereby improving training efficiency.
>
> ----
> > **Q7:** It is unclear/undefined what "guidance step size" in the top of page 5 means.
>
> **Response to Q7:**
>
> Thank you for pointing this out. We have clarified that the “guidance step size” refers to **the number of environment time steps at the beginning of each episode that are controlled by the prior policy before switching to the online policy**. In other words, it is the *step length* (in discrete time steps) of the prior-controlled prefix of each episode. We have updated the second paragraph of Sec. 5 accordingly.
>
> > **Q8:** The paper would be strengthened by an explicit explanation of what JSRL and QMIX do.
>
> **Response to Q8:** We have expanded the related-work section to briefly introduce both methods.
>
> QMIX: a value-factorization MARL algorithm that addresses the credit-assignment problem by learning per-agent local utility functions and combining them through a mixing network with non-negative weights generated by hypernetworks conditioned on the global state. The monotonicity constraint ensures that the global joint value is consistent with improvements in each agent’s local value.
>
> JSRL: a prior-guided framework in single-agent RL that focuses on incorporating prior policies into DRL training. It introduces a dual-policy architecture consisting of a guidance policy and an exploration policy: the guidance policy first drives the agent toward high-value regions, after which the exploration policy optimizes from those states. A curriculum that gradually reduces the guidance horizon mitigates reliance on the prior and alleviates the difficulty of initial exploration.
>
> > **Q9:** An explicit algorithm box would improve clarity, including which hyperparameters need to be tuned per-task.
>
> **Response to Q9:** Thank you very much for the valuable suggestion. We have improved the design of Fig. 3 (the overall schematic of DUCE) to more clearly illustrate the interaction between the two curricula, helping readers better understand the DUCE framework. In addition, Appendix A.3.2 provides a clear description of all hyperparameters related to DUCE; details are provided in our Response to W1.

---

### Official Review · Reviewer_6R9G · 2025-10-30

**Soundness:** 3
**Presentation:** 3
**Contribution:** 3
**Rating:** 6
**Confidence:** 2

**Summary:**

This paper proposes DUCE, which is curriculum on both task difficulty and policy optimization to enable a cold-start training in MARL.  In particular, DUCE utilizes similar curriculum on task difficulty inspired by JSRL and improves it to be adjusted by online performance feedback. Besides, DUCE also utilizes CQL loss to mitigate distribution shift and proposes to linearly decay the CQL loss with time $t$ for better balances between exploitation and exploration.

**Strengths:**

In spired from JSRL [1] for SARL, this paper proposes a cold-start O2O MARL algorithm, which is different from previous hot-start MARL methods.

This paper proposes a curriculum loss, which gradually degrades the weight of offline loss beyond JSRL.

The performance of the proposed method is much better than baselines on training efficiency and final results.

**Weaknesses:**

### Here are some concerns that may limit the contribution.
### 1. Confusion about Section 4 or Figure 2

1. What are the guidance policy and exploration policy in JSRL? It is important to show:
   1. The performance of the guidance policy $\pi^g$, which is trained well in offline dataset.
   2. The performance of the exploration policy $\pi^e$.
   3. The performance of the combined policy $\pi$.
2. What is the strategy for guide-step sequences, curriculum, and random-switching?
   1. Better to show both fails in MARL.
3. It is better to show that the guidance policy is significantly better than the exploration policy at the early stage, which is the central claim of JSRL.
4. Since JSRL originally belongs to single-agent RL, it is better to test how much benefit JSRL can provide to "6h vs 8z medium" by centralizing the agents to be single-agent before comparing with MARL.

### 2. Curriculum on task difficulty

1. The curriculum method for determining guide-step sequences is translated from JSRL, while the improvement of this method is insufficient. Here is the reason:
   1. The curriculum method in JSRL can be naturally applied in MARL.
   2. The only difference is that JSRL degrades the guide step based on a fixed rule, while this paper is based on experience feedback.
   3. The curriculum method in this paper doesn't consider any multi-agent property, like
      1. different agents may be suitable for different strategies for determining guide-step sequences.
      2. different agents may be suitable for different curriculum parameters.

### 3. Curriculum on offline loss

1. Utilizing offline CFCQL loss to mitigate the unlearning problem is first proposed in OVMSE [2].
2. The description from line 307 to line 309 seems to have a problem:
   1. Of course, OVMSE applies offline CFCQL loss in Offline Value Memory (OVM) for $Q_{OVM}$.
   2. However, $Q_{OVM}$ is applied to update $Q_{tot}$ partially weighted by $\lambda_{memory}$.
   3. Finally, the agent is updated via the joint action-value function $Q_{tot}$ in ft-QMIX.
   4. Thus, the policy is also influenced by offline loss, which is a similar idea in this paper.

3. Besides, OVMSE also gradually degrades the offline loss weight by annealing scheduling $\lambda_{memory}$, which is similar to the curriculum design on policy loss in this paper.
### 4. Baselines

The selected baselines are all RL-based methods. However, there is also another type of baseline on the generative model, such as Decision Transformer [3] (DT). This kind of method allows offline pretraining and online finetuning, which is O2O. Please compare with the work SO2-MADT [4], which is also an O2O work on MARL with code provided in the paper.

### 5. Scenario Selection

While the scenarios in SMAC v1 are comprehensive, the scenarios in SMAC v2 are similar. Is there any standard for the scenario selection beyond "widely recognized"? Could you please provide diverse tasks, such as different and asymmetric numbers of agents?

[1] Uchendu, I., Xiao, T., Lu, Y., Zhu, B., Yan, M., Simon, J., ... & Hausman, K. (2023, July). Jump-start reinforcement learning. In *International Conference on Machine Learning* (pp. 34556-34583). PMLR.

[2] Zhong, H., Wang, X., Li, Z., & Huang, L. (2024). Offline-to-Online Multi-Agent Reinforcement Learning with Offline Value Function Memory and Sequential Exploration. *arXiv preprint arXiv:2410.19450*.

[3] Chen, L., Lu, K., Rajeswaran, A., Lee, K., Grover, A., Laskin, M., ... & Mordatch, I. (2021). Decision transformer: Reinforcement learning via sequence modeling. *Advances in neural information processing systems*, *34*, 15084-15097.

[4] Shah, A. B., Wen, Y., Chen, J., Wu, X., & Fu, X. (2024, October). Safe Offline-to-Online Multi-Agent Decision Transformer: A Safety Conscious Sequence Modeling Approach. In *2024 IEEE/RSJ International Conference on Intelligent Robots and Systems (IROS)* (pp. 12400-12407). IEEE.

**Questions:**

Please refer to the Weaknesses part.

---

> ### Author Response · Authors · 2025-11-25
> **Response to Reviewer 6R9G (1/5)**
>
> We sincerely thank the reviewer for the careful reading and constructive comments.
>
> Below we respond to each concern and describe the changes we made to clarify the paper and strengthen the empirical evidence.
>
> ## Weaknesses
>
> ### Confusion about Section 4 / Figure 2
>
> > **W1:** What are the guidance policy and exploration policy in JSRL?  It is important to show ...
>
> **Response to W1:**  We apologize for the lack of clarity in the original draft. Following your suggestion, we have made the following revisions:
>
> 1. **Clarified the definitions in the related-work section**:
>    - The **guidance policy** $\pi_{guide}$ refers to the *prior policy*, which is frozen during online training and plays the role of a teacher guiding the learning process of the exploration policy.  For example, on the 6h_vs_8z medium-replay task, the prior policy is obtained by training CFCQL on the medium-replay offline dataset, achieving an average win rate of 5% and an average return of 13.33. The performance of prior policies for other tasks is reported in Table 4.
>    - The **exploration policy** $\pi_{explore}$ refers to the MARL policy that is learned online through trial and error.
>    - The **combined policy** denotes the policy formed by composing the guidance and exploration policies: the guidance policy interacts with the environment for a prefix of each episode, after which the exploration policy continues until termination.
>
> 2. We have added an experiment in Appendix A.4.4 illustrating the performance of the guidance policy, the exploration policy, and the combined policy in JSRL-CFCQL during preliminary runs on 6h_vs_8z-medium and 6h_vs_8z-medium-replay. In Fig. 12, the return curves clearly show that the guidance policy maintains constant performance throughout training and substantially outperforms $\pi_{\text{explore}}$ at initialization; in the early phase, the mixed policy remains close to the guidance policy, while the exploration policy gradually improves. When the curriculum completes and the guidance policy no longer interacts with the environment, the combined policy reduces to the exploration policy alone. After the curriculum, however, the conservative regularization from CFCQL imposes excessive penalties that constrain exploration, restrict the collected samples to a narrow local region of the state space, inflate in-distribution Q-values, and ultimately cause overfitting—highlighting that the core challenge in MARL lies in *how* prior guidance is incorporated.
>
> > **W2:** What is the strategy for guide-step sequences, curriculum, and random-switching? Better to show both fails in MARL.
>
> **Response to W2:**  In the preliminary experiments, both JSRL-QMIX and JSRL-CFCQL used JSRL’s curriculum-based scheme as the strategy for determining guide-step sequences. Following your suggestion, we added experiments that use **random-switching** as the strategy for guide-step sequences. Specifically, we designed **JSRL(Random)-CFCQL**, which replaces the curriculum in JSRL-CFCQL with random-switching, implemented exactly as in the original JSRL paper—that is, randomly sampling the number of guide-steps for every episode.
>
> The new results (added in Sec. 4) show that on 5m_vs_6m and 6h_vs_8z (medium and medium-replay), JSRL(Random)-CFCQL performs better than JSRL-CFCQL and does not suffer the sharp performance drop observed in the curriculum version. However, after 0.5M steps its improvement becomes very slow. This indicates that random switching increases sample diversity and alleviates the overly restrictive penalties of the conservative regularizer in the early phase, but because it does not adaptively reduce prior guidance as training progresses, later-stage performance remains constrained by the limited capability of the prior policy. Furthermore, both JSRL(Random)-CFCQL and JSRL-CFCQL perform significantly worse than ft-QMIX. This demonstrates that neither naive curricula nor random switching provide benefits in MARL, reinforcing our Observation I that a straightforward extension of JSRL to the MARL setting is insufficient.
>
> > **W3:** It is better to show that the guidance policy is significantly better  than the exploration policy at the early stage, which is the central  claim of JSRL.
>
>  **Response to W3:** To more clearly illustrate that the guidance policy significantly outperforms the exploration policy in the early stage, we have added an experiment in Appendix A.4.4 that explicitly reports the separate performance of $\pi_{guide}$ and $\pi_{explore}$ for JSRL-CFCQL in preliminary runs on *6h_vs_8z-medium* and *6h_vs_8z-medium-replay*. In Fig. 12, the return curves clearly show that  at the early phase, the guidance policy maintains constant performance throughout training and substantially outperforms $\pi_{\text{explore}}$  , and the exploration policy gradually improves. We believe this new experiment makes the JSRL “jump-start” intuition explicit.

---

> > ### Author Response · Authors · 2025-11-27
> >
> > Dear Reviewer 6R9G,
> >
> > As the author–reviewer discussion period will conclude soon, we would be grateful if you could kindly take a moment to look over our responses to your comments. If you have any additional questions or feedback, we would be very happy to address them while the discussion window is still open.
> >
> > Thank you very much for your time and consideration.
> >
> > Best regards,
> > The Authors

---

> ### Author Response · Authors · 2025-11-25
> **Response to Reviewer 6R9G (2/5)**
>
> > **W4:** Since JSRL originally belongs to single-agent RL, it is better to test  how much benefit JSRL can provide to "6h vs 8z medium" by centralizing  the agents to be single-agent before comparing with MARL.
>
> **Response to W4:**  This is an excellent suggestion; thank you.
>
> To isolate the effect of MARL-specific challenges (non-stationarity and credit assignment), we have added a **centralized single-agent experiment** in Appendix A.4.3. Specifically, we formulate a multi agent problem as a centralized control problem, where a single agent observes the global state and outputs the joint action. On top of this centralized formulation, we implement **IQL** and **JSRL-IQL**, i.e., applying JSRL to an IQL-style single-agent learner.
>
> The results (reported in Fig. 11) show that the reward curves clearly indicate in the early stage of training, JSRL-IQL exhibits a rapid performance increase compared with IQL, demonstrating that in the centralized setting, JSRL-IQL indeed provides a clear “jump-start” over the vanilla IQL baseline. This confirms that the JSRL curriculum is beneficial relative to no curriculum in the centralized formulation, which is precisely why we adopted JSRL as a promising starting point. However, centralized control suffers from the curse of dimensionality, making it difficult for JSRL-IQL to learn effective policies.
>
> When we move to the decentralized MARL setting (JSRL-QMIX), the same idea no longer yields improvements and even **hurts sample efficiency** compared with ft-QMIX—exactly as reported in Sec. 4.
>
> This comparison supports our main message: the core challenge does not lie in JSRL itself, but in **extending prior-guided cold-start training to factorized MARL under Dec-POMDPs**, which is precisely what DUCE is designed to address.
>
> ### Curriculum on task difficulty
>
> > **W5:**  The curriculum method for determining guide-step sequences is translated from JSRL, while the improvement of this method is insufficient. Here is the reason ...
>
> **Response to W5:**
>
> Thank you for this helpful comment. We clarify (i) how our curriculum differs from JSRL, (ii) the concrete effect of the two designs $p_{\text{guide}}$ and the performance-based schedule, and (iii) why we adopt a global (rather than per-agent) guidance schedule in our MARL setting.
>
> 1. **Can JSRL’s curriculum be “naturally” applied to MARL?**
>
>    We explicitly implemented JSRL on top of ft-QMIX, obtaining **JSRL-QMIX** and **JSRL-CFCQL** as naïve prior-guided cold-start MARL baselines (Sec. 4, Fig. 2). On 6h_vs_8z-medium / medium-replay, both variants show overall lower training efficiency than ft-QMIX and suffer from much larger $Q_{\text{tot}}$ and TD error, i.e., more severe over-estimation.  These negative results show that **simply “plugging JSRL into MARL” is not sufficient** in MARL. Motivated by these observations, we propose DUCE, a dual-track curriculum tailored to cold-start MARL with prior-guided exploration.
>
> 2. Our task-difficulty curriculum differs from JSRL in two tightly coupled mechanisms:
>
>    - **Interleaving guided vs. unguided episodes via $p_{\text{guide}}$.**
>       At the beginning of each episode we flip a Bernoulli variable with probability $p_{\text{guide}}$: with probability $p_{\text{guide}}$ we run a prior-guided mixed policy (prior for the first $h$ steps, then online policy), and with probability $1-p_{\text{guide}}$ we run *pure online exploration*. This design is specific to our MARL setting with an offline regularizer.
>
>       The ablations in Sec. 6.3 directly show this effect. Removing the task-based curriculum (DUCE-noGuide, which corresponds to effectively setting $p_{\text{guide}}=0$) makes the win-rate on 6h_vs_8z-medium drop below 10% and slows learning on Zerg 5v5-medium-replay (Fig. 5). Varying $p_{\text{guide}}\in{0.1,0.3,0.5,0.7,1.0}$ (Fig. 6a) shows that **moderate values $0.5\le p_{\text{guide}} <0.7 $** yield both fast and stable learning, while too small or too large values degrade early-stage efficiency.
>
>       Thus $p_{\text{guide}}$ is not a cosmetic change: it is crucial to balance prior-guided exploitation (using prior trajectories to make the offline loss effective) and online exploration (collecting diverse data unaffected by conservative penalties).
>
>    - **Performance-based schedule for the guidance horizon $h$ .**
>       In DUCE, we *shorten* $h$ to the next curriculum stage **only when** the mixed policy at the current stage reaches at least $\beta$ times the performance of the previous stage.
>       The sensitivity analysis in Fig. 6b shows that $\beta\in[0.9,1.0]$ leads to stable curves, whereas smaller $\beta$ causes early oscillations. Together with the ablation DUCE-noGuide vs. DUCE (Fig. 5), these results demonstrate that **the combination of $p_{\text{guide}}$ and the performance-based update of $h$ is essential for the improved training speed and stability of DUCE**.

---

> > ### Author Response · Authors · 2025-11-28
> >
> > Dear Reviewer 6R9G,
> >
> > As the author–reviewer discussion period will conclude soon, we would be grateful if you could kindly take a moment to look over our responses to your comments. If you have any additional questions or feedback, we would be very happy to address them while the discussion window is still open.
> >
> > Thank you very much for your time and consideration.
> >
> > Best regards, The Authors

---

> > > ### Comment · Reviewer_6R9G · 2025-11-28
> > >
> > > Thank you for your detailed response. I am willing to raise my score when it is available.

---

> ### Author Response · Authors · 2025-11-25
> **Response to Reviewer 6R9G (3/5)**
>
> **Response to W5:**
>
> 3. **Why not per-agent guide-step sequences and per-agent curriculum parameters?**
>
>    We agree that heterogeneous curricula are an interesting direction. In our experiments, however, all agents are share parameters, and are trained under CTDE with a factorized joint value $Q_{\text{tot}}$ (ft-QMIX). In this setting:
>
>    - The curriculum operates at the **episode level**, deciding for how long the *joint* prior policy interacts with the environment before *all* agents switch to the joint online policy.
>    - Allowing each agent to have its own guidance horizon and curriculum parameters would mix prior and online policies *within the same time step*, breaking the trajectory structure assumed by the centralized critic, exacerbating non-stationarity, and introducing many additional hyper-parameters without a clear inductive bias for homogeneous SMAC agents.
>
>    To better connect our per-agent ablations to your suggestions, we designed DUCE-G1 and DUCE-G2 as **minimal yet representative instantiations** of the two ideas:
>
>    > (i) different agents may be suitable for different strategies for determining guide-step sequences;
>    > (ii) different agents may be suitable for different curriculum parameters.
>
>    - In **DUCE-G1**, the first half of the agents follows a curriculum-based guidance horizon, whereas the second half samples the guide steps randomly at the episode level.
>    - In **DUCE-G2**, the first half follows a curriculum with 10 stages, while the second half follows one with 20 stages.
>
>    **Why DUCE-G1 mixes curriculum-based and random guide-step strategies.**
>    In JSRL, there are essentially **two canonical mechanisms** for controlling the guide-step sequence:
>    (a) a curriculum-based schedule that gradually shortens the guidance horizon; and
>    (b) a purely random switching strategy. DUCE-G1 therefore partitions the agents into two sub-groups, with one group following the curriculum-based guidance horizon and the other group sampling guide steps randomly at the episode level. This construction is intentionally simple: it directly realizes point (i) *without* introducing additional, ad-hoc strategies that do not appear in the original JSRL framework. In other words, DUCE-G1 is a **clean and interpretable test case** of “different agents follow different guide-step strategies”, built solely from the two standard mechanisms already present in JSRL. More sophisticated variants (e.g., many different strategies per agent or complex learned scheduling rules) would add a large number of design choices and hyper-parameters, making it harder to attribute any performance change to the core idea of per-agent strategy heterogeneity.
>
>    **Why DUCE-G2 varies the number of curriculum stages as the key curriculum parameter.**
>    For point (ii), we need a parameter that genuinely characterizes the curriculum, rather than a peripheral hyper-parameter. In our curriculum, the **number of stages** controls how finely the guidance horizon is annealed: more stages mean a finer-grained, slower decay of prior guidance, while fewer stages mean a coarser, more aggressive decay. We therefore let different sub-groups of agents share the same curriculum *structure* but use different numbers of stages (e.g., 10 vs. 20). This directly instantiates “different agents have different curriculum parameters” in a way that is both **semantically meaningful** (it changes the granularity of prior guidance) and **easy to interpret**. Again, we deliberately avoid an overly elaborate design (e.g., tuning many different parameters per agent), because our goal is to test the *principle* of agent-wise curriculum heterogeneity, not to engineer a highly optimized heterogeneous scheme.
>
>    The results in Fig. 21 of  Appendix A.5.9 show that, on 6h_vs_8z-medium and 6h_vs_8z-medium-replay, **both DUCE-G1 and DUCE-G2 exhibit slower early learning and slightly lower final win rates than the original DUCE with a single global curriculum**, and they do not bring systematic improvements to ft-QMIX. In other words, making the guidance schedule heterogeneous across agents tends to increase training variance without yielding consistent gains in our CTDE value-factorization setting.
>
>    Based on these observations, we chose a **single global guidance schedule**, which is already validated by our ablations as effective for improving MARL training efficiency. We will clarify this design rationale, the roles of $p_{\text{guide}}$ and $\beta$, and the newly added DUCE-G1 / DUCE-G2 results more explicitly in Sec. 5, Sec. 6.3, and Appendix A.5.9 of the revised manuscript.

---

> ### Author Response · Authors · 2025-11-25
> **Response to Reviewer 6R9G (4/5)**
>
> ### Curriculum on offline loss
>
> > **W6:**
> >
> > 1. Utilizing offline CFCQL loss to mitigate the unlearning problem is first proposed in OVMSE.
> > 2. The description from line 307 to line 309 seems to have a problem:
> >    1. Of course, OVMSE applies offline CFCQL loss in Offline Value Memory (OVM) for $Q_{\text{OVM}}$.
> >    2. However, $Q_{\text{OVM}}$ is applied to update $Q_{\text{tot}}$  partially weighted by $\lambda_{\text{memory}}$.
> >    3. Finally, the agent is updated via the joint action-value function $Q_{\text{tot}}$ in ft-QMIX.
> >    4. Thus, the policy is also influenced by offline loss, which is a similar idea in this paper.
> > 3. OVMSE also gradually degrades the offline loss weight by annealing scheduling $\lambda_{memory}$, which is similar to the curriculum design on policy loss in this paper.
>
> **Response to W6:** To our understanding, as stated in Sec. 4.3 of the OVMSE paper, it uses CQL rather than CFCQL to mitigate unlearning; In the Offline Value Memory (OVM) module, it constructs $Q_{\text{ovm}}$ from $Q_{\text{tot}}^{\text{offline}}$, which is trained in the offline stage using a CQL loss. The resulting $Q_{\text{ovm}}$ is then used to update $Q_{\text{tot}}$, partially weighted by $\lambda_{\text{memory}}$. Moreover, Sec. 5.1 of the OVMSE paper explicitly states that for complex SMAC tasks, no offline regularization term is used during online fine-tuning. For the reviewer's convenience, we copy the text from OVMSE as follows:
>
> “*for 3s5z, 5m_vs_6m, and 6h_vs_8z, we exclude the CQL loss because it hinders online learning by preventing the discovery of new strategies.*”
>
> In summary, OVMSE does not employ any offline loss in the online fine-tuning phase, and it does not use a linearly decaying schedule on the offline loss weight. This is fundamentally different from the offline regularization term designed in our policy-optimization curriculum.
>
> In addition, DUCE introduces a dual-track curriculum whose tight coordination markedly enhances cold-start MARL efficiency and performance. The two curricula interact via the replay buffer: under the task-difficulty curriculum, the mixed prior–online policy collects high-quality samples with a large fraction of prior-guided data; the policy-optimization curriculum then applies conservative updates to this mixed distribution, rapidly improving the online policy despite distributional shift. As the online policy strengthens and the task-difficulty curriculum advances, samples collected by the online policy increase and populate the replay buffer, after which the policy-optimization curriculum gradually transitions from conservative to exploration-driven updates. In essence, the two curricula operate synergistically: early training is dominated by prior-guided data and benefits from conservatism, whereas later training shifts toward online exploration as regularization diminishes.
>
> ### Baselines
>
> > **W7:** The selected baselines are all RL-based methods. However, there is also  another type of baseline on the generative model, such as Decision  Transformer (DT). This kind of method allows offline pretraining and online finetuning, which is O2O. Please compare with the work SO2-MADT, which is also an O2O work on MARL with code provided in the paper.
>
> **Response to W7:** We agree that O2O methods based on generative models are highly relevant and thank you for pointing this out. We have discussed SO2-MADT in the related-work section. SO2-MADT leverages transformer architectures, which are particularly well suited for modeling long-horizon temporal dependencies and complex multi-agent interactions. However, as shown in Fig. 3 of the SO2-MADT paper, on maps such as MMM2, SO2-MADT is reported to require up to 10M steps to reach high win rates, whereas DUCE attains comparable performance in fewer than 3M steps. Although the paper does not explicitly describe the quality of the offline datasets used, an inspection of the released code indicates that lines 30–31 of main_so2_madt.py default to the *good* dataset level (according to the MADT appendix, *good* datasets are collected by near-optimal policies and are therefore of high quality). By contrast, DUCE, as described in Section 6.1, uses only *medium-quality* datasets to obtain suboptimal prior policies for guidance. Taken together, these observations suggest that DUCE is improved sample efficiency compared to SO2-MADT under weaker prior data assumptions.

---

> ### Author Response · Authors · 2025-11-25
> **Response to Reviewer 6R9G (5/5)**
>
> ### Scenario Selection
>
> > **W8:** While the scenarios in SMAC v1 are comprehensive, the scenarios in SMAC  v2 are similar. Is there any standard for the scenario selection beyond  "widely recognized"? Could you please provide diverse tasks, such as  different and asymmetric numbers of agents?
>
> **Response to W8:**  We appreciate the request for greater diversity and have clarified our selection criteria while extending the experiments accordingly.
>
> 1. **Selection criteria**.
>
>    Our original benchmark suite was chosen to cover:
>
>    - **Different difficulty levels:**
>
>      **5m_vs_6m** (hard) vs. **6h_vs_8z**, **Corridor**, **MMM2** (super-hard).
>
>    - **Different and asymmetric agent numbers in SMAC v1:**
>
>      *5m_vs_6m* and *6h_vs_8z* correspond to naturally asymmetric settings (5 vs. 6 and 6 vs. 8), exactly matching the reviewer’s request for unbalanced team sizes.
>      *Corridor* and *MMM2* exhibit even stronger imbalance and asymmetry.
>
>    - **Different unit-type configurations in SMAC v2**
>
>      *Protoss_5v5* and *Zerg_5v5* randomly generate the number of units of each type at the start of every episode, unlike SMAC v1 which uses fixed unit compositions.
>
> 2. **Continuous-control benchmark**
>
>    We have added experiments in Appendix A.5.3 evaluating DUCE on the widely recognized **MA-MuJoCo** benchmark to demonstrate its applicability to continuous-action multi-agent settings.
>
>    1. **Baselines.**
>       Following our findings on SMAC v1/v2, where ft-QMIX already served as a challenging baseline, we now move to continuous-action MARL settings. Since ft-QMIX cannot be directly applied to continuous action spaces, we select **MATD3**, a widely used actor–critic method for continuous cooperative MARL, as a strong baseline, together with **OVMSE**. Because OVMSE has no official implementation, we re-implemented OVM and SE based on the MATD3 code in the **HARL** repository; all other implementation details follow Sec. 6.1. For fairness, DUCE uses the *same prior policy* as that produced by OVMSE during its offline stage. To adapt DUCE to continuous action spaces, we adopt **MATD3** as the underlying MARL algorithm, using its implementation from HARL.
>
>    2. **Experimental setup.**
>       We additionally conducted experiments on the widely recognized **Multi-agent MuJoCo (MA-MuJoCo)** benchmark, including challenging scenarios: *HalfCheetah-v2 3×2*, *HalfCheetah-v2 2×3* and *HalfCheetah-v2 6×1*. Each scenario has two dataset tasks—*medium* and *medium-replay*.
>       Following the procedure described in the CFCQL paper, datasets were constructed using the **MATD3** algorithm: the medium datasets contain 1000 trajectories, while the medium-replay datasets contain no more than 1000 trajectories(full details in Table 3 and  Appendix A.5.3).
>
> ​As shown in Fig. 15 of Appendix A.5.3, **DUCE markedly outperforms MATD3 and OVMSE**, converging faster and achieving higher asymptotic performance, even though OVMSE benefits from warm starts that initially give it an advantage.
>
> ​The HARL repository used in our implementation is available at: [github](https://github.com/PKU-MARL/HARL)

---

### Meta-Review · Area_Chair_mk4T · 2025-12-30

**Summary:**

This paper propose DUCE, a prior-guided cold-start MARL algorithm with suboptimal priors, aiming to balance exploitation and exploration to ensure efficient, stable cold-start training. The curriculum designs include: (1) an externally configured task-difficulty curriculum that alternates between performing prior and online policies with probabilistic scheduling, and (2) an internally evolving policy optimization curriculum that imposes a decaying offline RL regularizer on the online loss, enabling a smooth shift from conservative prior reliance to exploration-driven training.  DUCE is built upon JSRL and utilizes similar curriculum on task difficulty inspired by JSRL, and it further improves it in multi-agent settings, by using online performance feedback.

**Reviewer Concerns:**

The reviewers raises a few major concerns:  1) Limited novelty beyond JSRL:  A few reviewers stated this paper used the two key modifications (beyond JSRL and CFCQL) in the literature, but it  did not present sufficient  concrete analysis on the benefits of  cross-interaction of these two ideas.  In the rebuttal, the authors made efforts to address this concer, but more work is needed to make a more convincing case. 2) Additional  computational complexity incurred by optimizing two new key hyperparameters, i.e.  the regularization weight and the length of the decay period, besides  other hyperparameters inherited from prior work.

**Reviewer Scores:**

The review scores are /6/2/6/2

---

### Decision · Program_Chairs · 2026-01-26

Reject